# An integrated platinum-nanocarbon electrocatalyst for efficient oxygen reduction

Lei Huang [1], Min Wei[2], Ruijuan Qi [3], Chung-Li Dong [4], Dai Dang[5], Cheng-Chieh Yang[4], Chenfeng Xia[1], Chao Chen[5], Shahid Zaman [1], Fu-Min Li[1], Bo You [1] & Bao Yu Xia [1] ✉

Efficient and robust platinum-carbon electrocatalysts are of great significance for the long-term service of high-performance fuel cells. Here, we report a Pt alloy integrated in a cobalt-nitrogen-nanocarbon matrix by a multiscale design principle for efficient oxygen reduction reaction. This Pt integrated catalyst demonstrates an increased mass activity, 11.7 times higher than that of commercial Pt catalyst, and retains a stability of 98.7% after 30,000 potential cycles. Additionally, this integrated catalyst delivers a current density of $1.50\,A\,cm^{-2}$ at 0.6 V in the hydrogen-air fuel cell and achieves a power density of $980\,mW\,cm^{-2}$. Comprehensive investigations demonstrate that the synergistic contribution of components and structure in the platinum-carbon integrated catalyst is responsible for the high-efficiency ORR in fuel cells.

High-efficiency catalysts used in the oxygen reduction reaction (ORR) essentially determine the service level and operating life of fuel cells, which is a promising energy technology for the sustainable development of human society[1]. Currently, platinum (Pt)-based components are essential and still irreplaceable for improving sluggish cathodic ORR kinetics[2–4]. Consequently, various strategies have been developed to improve the performance of Pt-based catalysts[5–7]. Generally, optimizing the crystal phase, strain effect and electronic distribution, interface adsorption would enhance the intrinsic activity, while tailoring sizes, morphologies and nanostructures would increase the utilization of active sites[8–11]. As a result, some nanostructured Pt-based electrocatalysts exhibit excellent catalytic activities in promoting the ORR, at least at the rotating disk electrode (RDE) level. Nevertheless, it has been well documented that very few RDE-level activities can be well transformed into membrane electrode assembly (MEA)[12]. Such activity inconsistency is attributed to the different reaction interfaces and operating conditions of the catalyst in RDE and MEA, which has become a critical limitation for the scale-up application of Pt-based catalysts in fuel cells[13–15].

In the construction of the catalyst layers and the assembly of MEA, the employment of carbon supports is usually indispensable, in which they ensure the dispersion and prevent the agglomeration of active Pt nanoparticles, especially the formation of pores and channels, to meet the requirements of sufficient electron transfer, mass transport, and product release. Therefore, integrating Pt alloys into the graphitized nanocarbons with continuous network and hierarchical architecture is expected to meet the requirements of the gas–liquid–solid reaction microenvironment and help Pt catalysts express activity and maintain stability in MEA. In addition, the metal-nitrogen-carbon motifs in the integrated hybrid provide additional active sites and strong interactions, thus synergistically stimulating higher activity and stability of the resultant Pt-based catalysts[16–18]. Consequently, the multiscale design, from component and structure, is committed to enhancing the high-level service of ORR catalysis by constructing the optimal gas–liquid–solid three-phase interface while ensuring the complete transformation from activity to performance.

[1]School of Chemistry and Chemical Engineering, Key Laboratory of Material Chemistry for Energy Conversion and Storage (Ministry of Education), Hubei Key Laboratory of Material Chemistry and Service Failure, Hubei Engineering Research Center for Biomaterials and Medical Protective Materials, Wuhan National Laboratory for Optoelectronics, Huazhong University of Science and Technology (HUST), 1037 Luoyu Rd, Wuhan 430074, China. [2]The Institute for Advanced Studies, Wuhan University, 299 Bayi Rd, Wuhan 430072, China. [3]Key Laboratory of Polar Materials and Devices (MOE), Department of Electronics, East China Normal University, Shanghai 200241, China. [4]Department of Physics, Tamkang University, 151 Yingzhuan Road, New Taipei City 25137 Taiwan, China. [5]School of Chemical Engineering and Light Industry, Guangdong University of Technology, Guangzhou 510006, China. ✉e-mail: byxia@hust.edu.cn

Here, we report an efficient platinum-nanocarbon integrated electrocatalyst for the ORR process in fuel cells. The hybrid catalyst is realized by multiscale design from the architecture construction of nanocarbon supports and atomic-level component adjustment of Pt alloy and cobalt-nitrogen-carbon (Co-N-C). The resultant PtCo@CoNC/NCNT/rGO (PtCo@CoNC/NTG) catalyst shows a high mass activity of 1.52 A mg$_{Pt}^{-1}$ at 0.9 V versus reversible hydrogen electrode (vs. RHE), which is 11.7 times higher than that of commercial Pt/C and only declined by 1.3% after 30,000 potential cycles. Additionally, this integrated catalyst delivers a current density of 1.50 A cm$^{-2}$ at a voltage of 0.6 V in a hydrogen-air fuel cell and reaches a power density of 980 mW cm$^{-2}$. Comprehensive investigations reveal that such excellent performance expression at RDE and MEA would be attributed to the collaborative contribution of multiple kinds of active sites and hierarchical nanocarbon matrix in PtCo@CoNC/NTG, which not only improves the utilization of active sites but also strengthens the electron transfer and mass exchange to jointly promote the ORR process. This work of Pt electrocatalysts may evoke profound research on multiscale design for integrated electrocatalysts in fuel cells and beyond.

## Results

### Material synthesis and structural characterizations

Figure 1 illustrates the design principle in the structure and components of three-dimensional (3D) PtCo@CoNC/NTG architectures. The Pt species are first attached to the reduced graphene oxide (rGO), and then the zeolitic-imidazolate frameworks (ZIFs) are deposited on the as-prepared Pt/rGO to obtain Pt@ZIFs/rGO (Supplementary Fig. 1). After pyrolysis treatment in a reducing environment, nitrogen-doped carbon nanotubes (NCNTs) were generated, and the resultant PtCo@CoNC/NTG with a hierarchical architecture was obtained (Supplementary Fig. 2)[19]. Following a similar concept, different dimensional PtCo@CoNC integrated catalysts are designed as PtCo@CoNC, PtCo@CoNC/CNT and PtCo@CoNC/rGO (Supplementary Fig. 3). In this preparation process, the nanocarbon construction and atomic-level component adjustment of active Pt-Co alloy and Co-N-C sites are simultaneously integrated into a hybrid architecture, which promotes local mass transport and electron transfer and thus collectively strengthens the synergistic ORR process[20].

The scanning electron microscopy (SEM) image first shows twisted nanosheets covered with a large amount of carbon species

(Fig. 2a), and a closer observation reveals dense NCNTs planted on the surface of the nanosheets (Fig. 2b). Moreover, the high-angle annular dark field-scanning transmission electron microscopy (HAADF-STEM) pattern demonstrates the continuous 3D network of the resultant PtCo@CoNC, which is mainly composed of a thin 2D layer of rGO, abundant 1D continuous NCNTs and 0D Pt-Co alloy nanoparticles (Fig. 2c). Notably, Pt-Co nanoparticles with sizes of 3-5 nm are wrapped by 2-3 carbon layers and uniformly planted in the matrix (Fig. 2d)[21]. The clear lattice spacing of the carbon layers (0.34 nm) and Pt alloy (0.22 nm) can be observed in a high-resolution (HR)TEM image, which corresponds to graphite carbon (002) and PtCo(111) crystal planes, respectively (Fig. 2e). The decrease of Pt lattice space indicates the contraction stress in the alloy nanoparticles, which would be caused by the introduction of smaller cobalt atoms. These features suggest that the robust structure of the PtCo@CoNC/NTG hybrid would help to inhibit Ostwald ripening and enhance the anticorrosion ability of Pt alloy nanoparticles while simultaneously stimulating synergistic interactions to promote ORR electrocatalysis (Supplementary Fig. 4a, b)[22]. Moreover, the aberration corrected (AC)-HAADF-STEM image of PtCo@CoNC/NTG reveals that abundant single atoms are uniformly dispersed in the modified graphitic carbon matrix (Fig. 2f). X-ray energy dispersive spectroscopy (EDS) mapping shows the element distribution and atomic ratio (Pt/Co = 41/59) of PtCo@CoNC/NTG, in which C and N elements are evenly distributed in the whole nanocarbon architecture, Pt species are mainly concentrated on the entire Pt-Co nanoparticles, while Co atoms are mostly distributed in the alloy and a partial proportion of atoms are dispersed in the carbon matrix (Fig. 2g, Supplementary Fig. 4c, d). Accordingly, the electron energy-loss spectrum (EELS) analysis of the isolated single-metal bright spots verifies the coexistence of C, N and Co atoms in the Co-N-C configuration (Supplementary Fig. 5). The line-scan profile clarifies the uniform distribution of Pt and Co species in the nanoparticles (Supplementary Fig. 6). Based on the microstructure analysis of different PtCo@CoNC samples (Supplementary Fig. 7), the multiscale design of integrated PtCo@CoNC catalysts is successfully elucidated in the dimensional microstructure and atomic-level component (Supplementary Fig. 8), which is essential to the critical issues in the ORR process, including the full utilization of multiple active sites, optimized mass diffusion and exchange, and accelerated electron transfer[23–25].

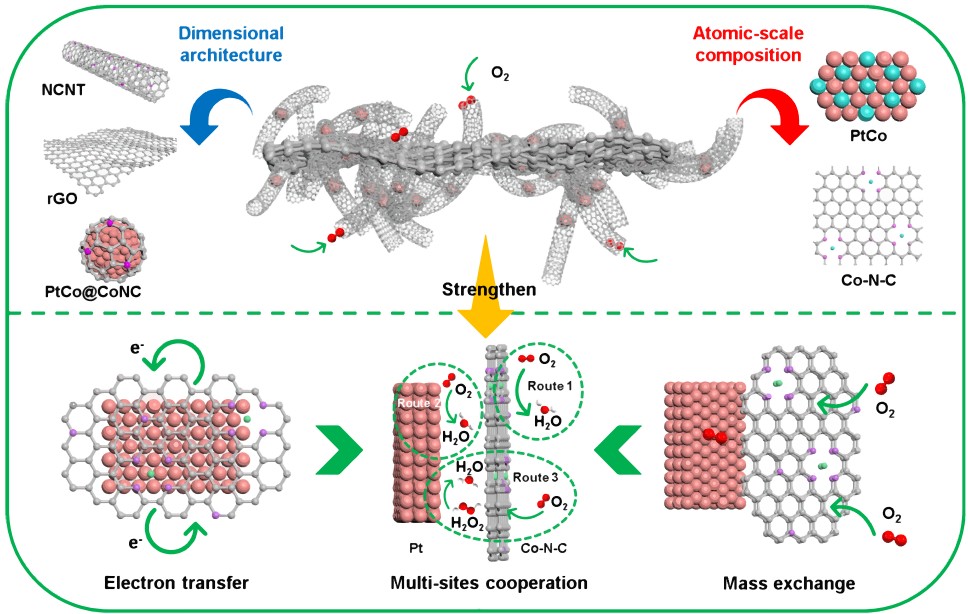

**Fig. 1 | Schematic diagram of the improvement mechanism for PtCo@CoNC/NTG.** Multiscale design illustration of alloyed platinum-nanocarbon integrated electrocatalysts for strengthening the ORR process.

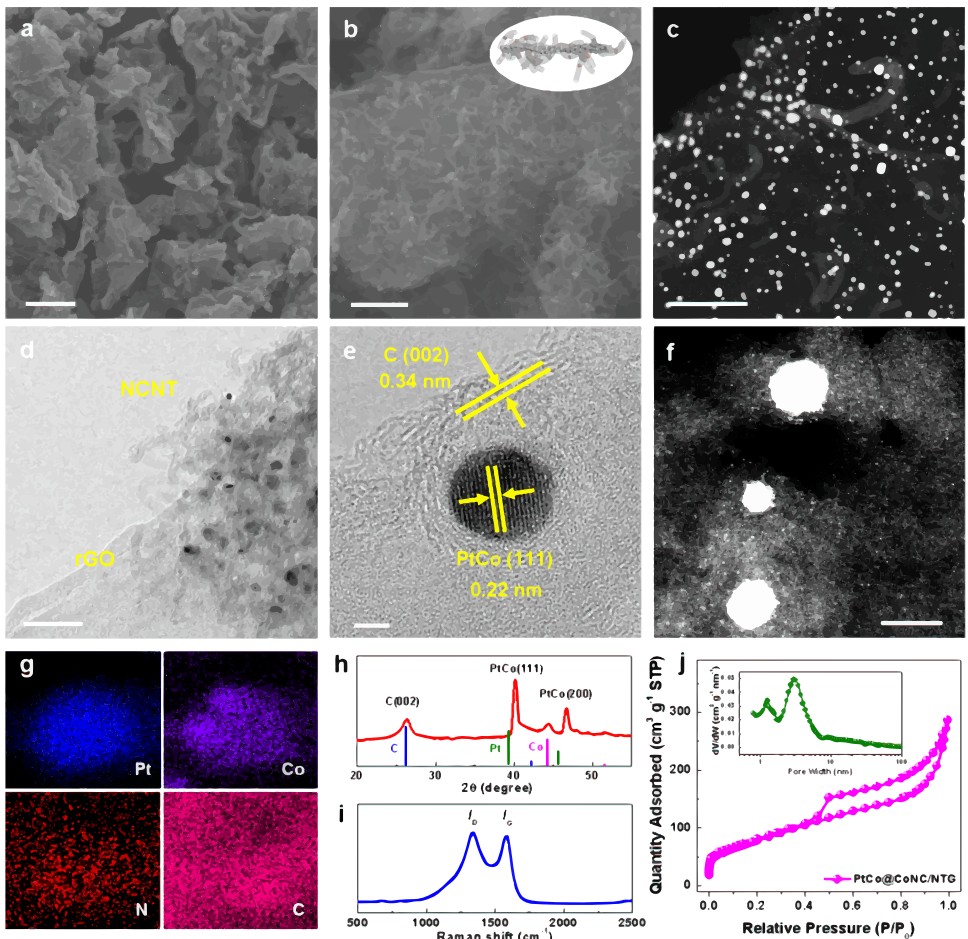

**Fig. 2 | Structure characterizations of PtCo@CoNC/NTG. a, b** SEM images, **c** HAADF-STEM image, **d** TEM image, **e** HRTEM image, **f** AC-HAADF-STEM image, **g** EDS mapping, **h** XRD pattern, **i** Raman spectrum, and **j** N$_2$ sorption curves of PtCo@CoNC/NTG, inset **j** is pore size distribution. The scale bars in **a, b, c, d, e** and **f** are 2 μm, 200 nm, 100 nm, 50 nm, 2 nm and 5 nm, respectively.

The crystal structure of PtCo@CoNC/NTG was examined by X-ray powder diffraction (XRD) analysis, and the diffraction peaks at 26.2°, 40.2°, 44.2° and 46.6° were assigned to the C(002), PtCo(111), Co(111) and PtCo(200) crystal planes, respectively (Fig. 2h). Compared with commercial Pt/C (Supplementary Fig. 9), a positive peak shift at 40.2° verifies the generation of contraction strain in the Pt-Co alloy, which is believed to enhance the ORR activity and reduce Pt consumption[26–28]. The graphitization information of the carbon components in the PtCo@CoNC integrated catalysts was studied by Raman spectroscopy (Fig. 2i). A larger value of $I_D/I_G$ (2.90) for PtCo@CoNC/NTG than PtCo@CoNC (1.66), PtCo@CoNC/CNT (1.98) and PtCo@CoNC/rGO (2.61) suggests more defective sites in the nanocarbon matrix (Supplementary Fig. 10). Compared with other PtCo@CoNC catalysts, the continuously constructed PtCo@CoNC/NTG exhibits a higher specific surface area of 297.9 cm$^3$ g$^{-1}$ (Fig. 2j), which would facilitate the widespread exposure of active sites and comprehensive contact with reactive species (Supplementary Fig. 11a)[29]. The pore size distribution of PtCo@CoNC/NTG is dominated by mesopores with a size of ~3 nm (Fig. 2j inset), and its mesopore ratio is significantly greater than that of other PtCo@CoNC hybrids (Supplementary Fig. 11b)[30]. Therefore, the balance of graphitic degree and porosity for the PtCo@CoNC/NTG integrated catalyst grasps better conductivity, stronger corrosion resistance and denser catalytic active sites, which would update the construction of a robust three-phase interface of MEA to promote active site exposure, improve the reaction kinetics and enhance electrochemical stability[31].

## Chemical state and coordination analysis

X-ray photoelectron spectroscopy (XPS) was then employed to examine the surface chemistry[32]. The full XPS survey mainly displays Pt 4$f$, Co 2$p$, C 1$s$, N 1$s$ and O 1$s$ signals, which perfectly correspond to the types of elements contained in PtCo@CoNC/NTG (Supplementary Fig. 12). The characteristic peak C 1$s$ is fitted into C-C (284.7 eV), C-N/C-O (285.7 eV) and C = O (290.8 eV) peaks, suggesting possible nitrogen doping in the hybrid matrix. Meanwhile, the N 1$s$ spectrum is deconvoluted into pyridinic N (398.5 eV), Co-N$_x$ (399.5 eV), pyrrolic N (401.0 eV) and graphitic N (402.6 eV), further confirming the presence of Co-N-C sites (Fig. 3a)[33]. The presence of Co in the form of Co 2$p_{3/2}$ and Co 2$p_{1/2}$ involves Co(0) 2$p_{3/2}$ (778.7 eV) and 2$p_{1/2}$ (793.9 eV), Co(II/III) 2$p_{3/2}$ (781.9 eV) and 2$p_{1/2}$ (797.3 eV), and a pair of satellite peaks, which are attributed to the Pt-Co alloy, atomic Co species in the Co-N-C configuration and some oxidized Co species (Fig. 3b). Based on the peak fitting of Pt(0) 4$f_{7/2}$ (71.5 eV), Pt(II) 4$f_{7/2}$ (72.6 eV), Pt(0) 4$f_{5/2}$ (74.9 eV) and Pt(II) 4$f_{5/2}$ (76.3 eV), it is further demonstrated that Pt is mainly in the metallic form and has a small amount of coordination state with oxygen, carbon or nitrogen (Fig. 3c). Compared with standard Co 2$p_{3/2}$ (778.3 eV) and Pt 4$f_{7/2}$ of Pt/C (71.7 eV), the decrease of Pt and the increase of Co in the binding energy indicate the electron transfer from Co to Pt (Supplementary Fig. 13). The above XPS analysis results primarily reveal the electronic interaction between different metals and Co-N-C caused by electron migration.

The local coordination and electronic structures were further probed by X-ray adsorption spectroscopy (XAS). Pt in PtCo@CoNC/

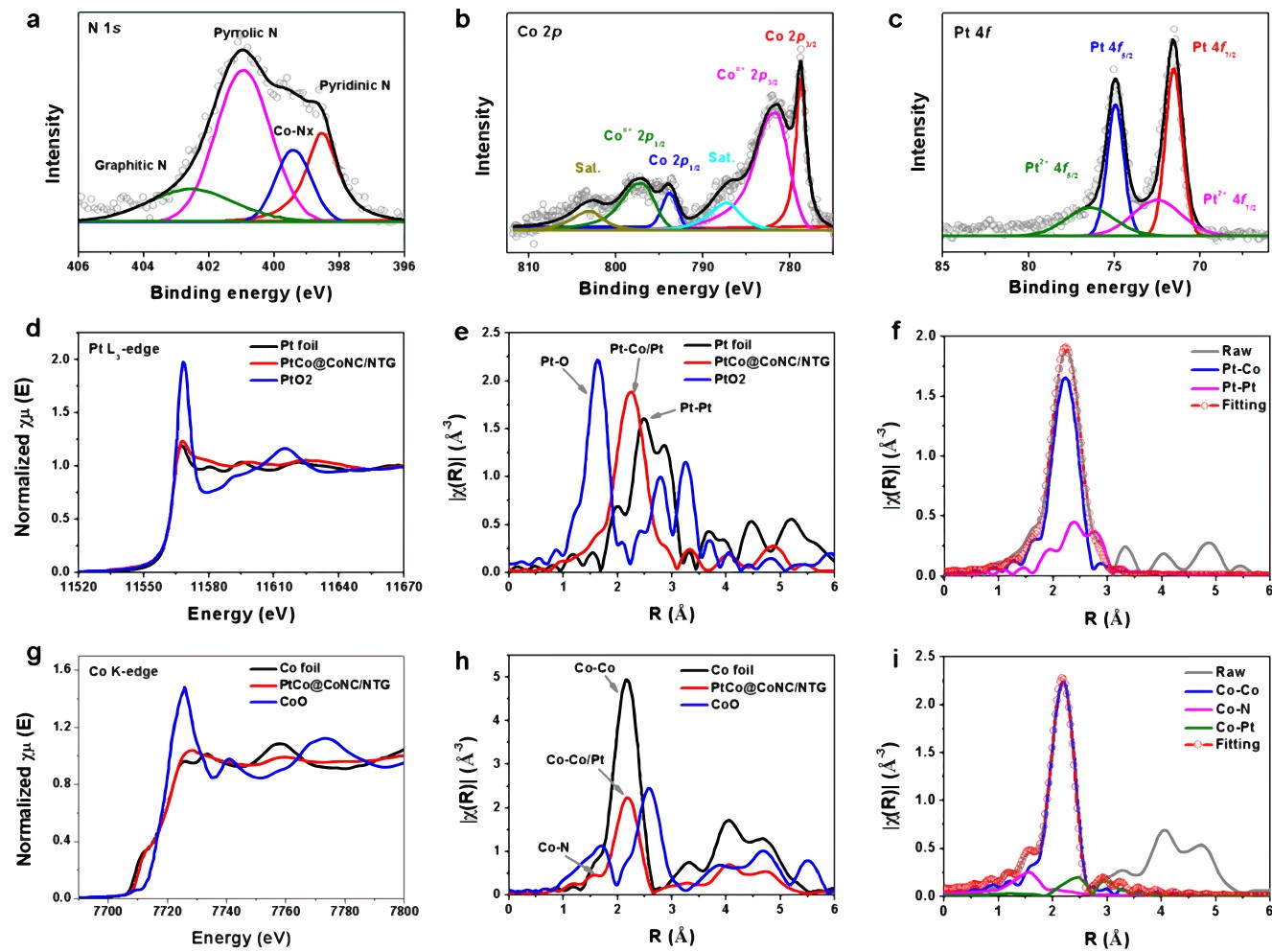

**Fig. 3 | Chemical state and coordination environment of PtCo@CoNC/NTG.**
**a–c** XPS spectra of N 1s (**a**), Co 2p (**b**) and Pt 4f (**c**). **d** Pt L$_3$-edge XANES, **e** the Fourier transforms of EXAFS spectra, and **f** the applied scattering paths of Pt L$_3$-edge for PtCo@CoNC/NTG. **g** Co K-edge XANES, **h** the Fourier transforms of EXAFS spectra, and **i** the applied scattering paths of Co K-edge for PtCo@CoNC/NTG.

NTG shows the metallic form as seen from the similar white line intensity (~11,568 eV) with Pt foil in X-ray absorption near-edge structure (XANES) spectra, which matches well with XPS results (Fig. 3d)[34]. Similarly, the Fourier transformed extended X-ray absorption fine structure (FT-EXAFS) for PtCo@CoNC/NTG at Pt L$_3$-edge shows only one scattering peak at ~2.2 Å, assigned to the contribution of Pt-Pt/Pt-Co bonds, which is smaller than that for Pt foil (Fig. 3e). Due to heteroatomic interactions in Pt-Co alloying, a shortened Pt-Pt distance is displayed compared to the standard Pt foil, which is perfectly consistent with the fitted Pt-Pt (~2.2 Å) and Pt-Co (~2.4 Å) signals (Fig. 3f)[35]. Compared with the k and R space diagrams of Pt foil and PtO$_2$ (Supplementary Fig. 14), the fitting information presented by PtCo@CoNC/NTG further supports a compressive Pt-Pt distance and an optimized coordination environment (Supplementary Table 1). In contrast, the white line intensity of the Co K-edge in PtCo@CoNC/NTG (~7726 eV) is located between Co foil and CoO, suggesting partial oxidation of Co (Fig. 3g). AC-STEM observations of Pt-Co nanoparticles and Co single atoms simultaneously existing in PtCo@CoNC/NTG suggest that the oxidized Co species are contributed by Co single atoms (Fig. 3h)[36]. Moreover, FT-EXAFS at the Co K-edge shows a main peak at ~2.2 Å and some shoulder peaks at ~1.5 and ~2.5 Å (Fig. 3i, Supplementary Fig. 15), which reasonably fit with the scattering paths of Co-Co, Co-N, and Co-Pt, respectively (Supplementary Table 2). This structural information supports that the Co-N-C moieties and Pt-Co alloy sites in PtCo@CoNC/NTG simultaneously act as ORR active sites[37–39]. The XAS

spectrum analysis also shows that other PtCo@CoNC integrated catalysts possess similar chemical components, including Pt-Co alloy and Co-N-C configuration (Supplementary Fig. 16, 17). Therefore, the 3D carbon network in the PtCo@CoNC/NTG hybrid is distinctive compared to other PtCo@CoNC samples, which is conducive to smooth mass transport and continuous electron transfer.

## ORR performance evaluation

Their electrochemical activities are then evaluated using linear sweep voltammetry (LSV) technique (Fig. 4a). PtCo@CoNC/NTG shows a much higher half-wave potential of 0.94 V vs. RHE, compared with PtCo@CoNC/rGO (0.89 V), PtCo@CoNC/CNT (0.88 V), PtCo@CoNC (0.86 V) and commercial Pt/C (0.87 V). Moreover, PtCo@CoNC/NTG exhibits a lower Tafel slope of 71 mV dec$^{-1}$ than that of the other samples, suggesting enhanced ORR kinetics (Fig. 4b). The PtCo@CoNC/NTG catalyst also delivers a significantly strengthening mass activity of 1.52 A mg$_{Pt}^{-1}$ at a potential of 0.9 V vs. RHE (Supplementary Table 3) compared with PtCo@CoNC/rGO (0.86 A mg$_{Pt}^{-1}$), PtCo@CoNC/CNT (0.42 A mg$_{Pt}^{-1}$), and PtCo@CoNC (0.25 A mg$_{Pt}^{-1}$), which is even the 11.7-fold enhancement compared to 0.13 A mg$_{Pt}^{-1}$ for commercial Pt/C (Fig. 4c). The different catalytic activities of these four PtCo@CoNC samples are mainly attributed to the differences in carbon specific surface area, pore size distribution, and carbon composition verified above[40–42]. Moreover, an accelerated durability test (ADT) of 30,000 cycles is measured to evaluate the electrochemical

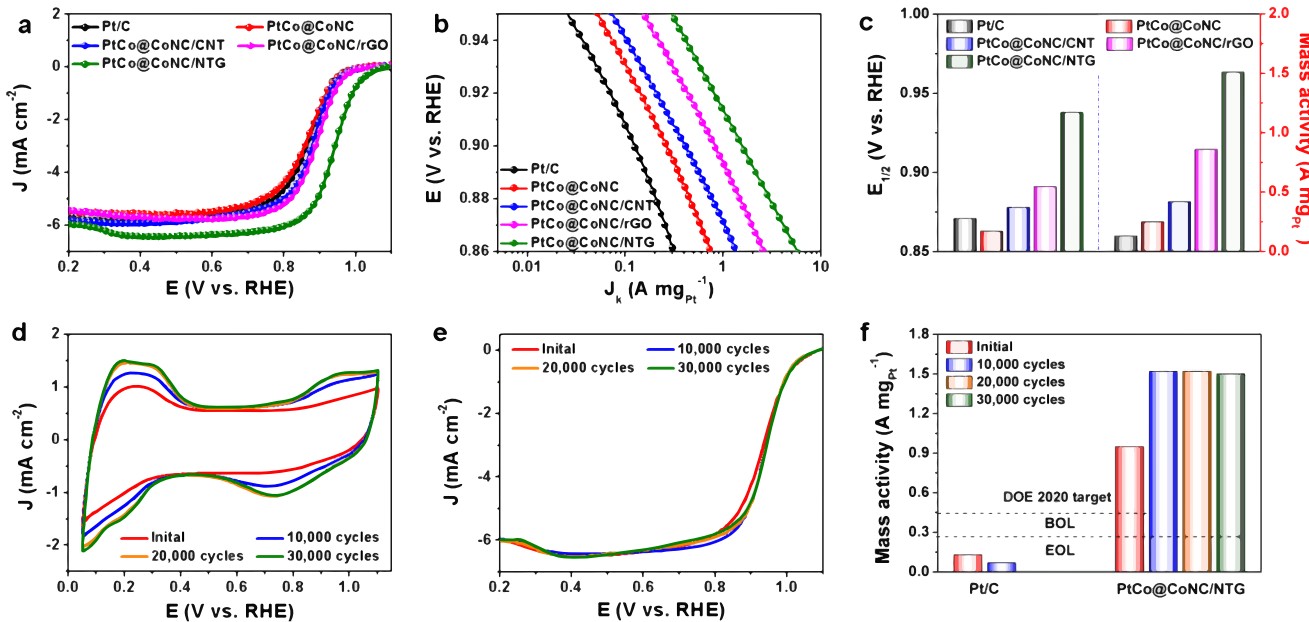

**Fig. 4 | Electrochemical evaluation of PtCo@CoNC/NTG. a** LSV curves, **b** Tafel plots, **c** half-wave potential and mass activity at 0.9 V vs. RHE of different Pt catalysts. **d** CV profiles, **e** LSV curves, and **f** activity comparison before and after ADTs of Pt/C and PtCo@CoNC/NTG.

stability of the PtCo@CoNC/NTG catalyst. Cyclic voltammetry (CV) and the corresponding absorption/desorption areas of hydrogen and oxygen expand to a stable state before and after 30,000 cycles, implying that the active sites at the catalytic interface are improved during long-term activation and then stabilized (Fig. 4d). Meanwhile, the limiting current platform of the LSV curve increases in the first 10,000 cycles and then remains steady in the subsequent 20,000 cycles (Fig. 4e). Correspondingly, the beginning-of-life (BOL) activity reaches 1.52 A mg$_{Pt}^{-1}$ at the 10,000th cycle, and the end-of-life (EOL) is attenuated by only 1.3% in the consequent 20,000 cycles (Fig. 4f). Compared with the Pt/C catalyst, which retains only 54% of its activity after 10,000 cycles, the activity decline of PtCo@CoNC/NTG is negligible. In contrast to using pure carbon as a support, the stability improvement of Pt catalysts by constructing a graphitic nanocarbon network is remarkable. The commercial Pt/C catalyst displays significant particle agglomeration and active site reduction after the potential cycle sweep, while the primary structure of PtCo@CoNC/NTG is well maintained, suggesting that the integration of graphitic nanocarbon and active Pt-Co alloy endows its robust spatial structure and stable electrochemical performance (Supplementary Fig. 18).

RDE measurement environment usually involves forced convection and diffusion of delicately thin catalyst layer on the electrode. When RDE activity is converted into MEA performance, it is difficult to evaluate the efficient mass transport and full utilization of active sites at the three-phase interface[43]. To further verify the ORR performance recorded at the RDE level and bridge the gap between the RDE and MEA evaluation, gas diffusion electrode (GDE) measurements were performed to clarify the gas transport efficiency (Supplementary Fig. 19a, b)[44]. It is worth noting that the starting potentials of PtCo@CoNC/NTG, Pt/C, PtCo@CoNC/rGO, PtCo@CoNC/CNT and PtCo@CoNC at 1 mA cm$^{-2}$ are 0.87, 0.87, 0.86, 0.84 and 0.80 V (Fig. 5a inset), while they drive the current density of 100 mA cm$^{-2}$ at 0.47, 0.40, 0.39, 0.33 and 0.31 V, respectively, indicating that PtCo@CoNC/NTG has faster electrochemical reaction efficiency and enhanced oxygen diffusion strength under the same conditions (Fig. 5a)[45]. Moreover, compared with the continuous degradation of Pt/C, the polarization current of PtCo@CoNC/NTG first increases and then remains unchanged for

300 potential cycles, which is similar to the change trend assessed by RDE (Fig. 5b). PtCo@CoNC/NTG also shows negligible current fluctuation in the long-term durability test at 0.6 V vs. RHE for >10 h (Fig. 5c)[46]. GDE evaluation builds a bridge between laboratory-level RDE screening and MEA evaluation, which is beneficial for quickly screening out high-efficiency ORR catalysts that are in line with the actual operation of fuel cells (Supplementary Fig. 20). Inspired by the principles of integrated engineering and multiscale design, we then investigate their translation of RDE activity to MEA performance (Supplementary Fig. 19c, d)[47]. The polarization plots of Pt/C and PtCo@CoNC cathode catalysts in hydrogen-air fuel cells are shown in Fig. 5d. PtCo@CoNC/NTG delivers a current density of 1.50 A cm$^{-2}$ at 0.6 V$_{iR-uncorrected}$ and a maximum power density of 980 mW cm$^{-2}$, outperforming PtCo@CoNC/rGO (1.04 A cm$^{-2}$, 850 mW cm$^{-2}$), PtCo@CoNC/CNT (1.03 A cm$^{-2}$, 820 mW cm$^{-2}$), PtCo@CoNC (0.84 A cm$^{-2}$, 680 mW cm$^{-2}$), and commercial Pt/C (1.21 A cm$^{-2}$, 780 mW cm$^{-2}$) catalysts. Noticeably, the polarization current at 800 mV$_{iR-uncorrected}$ is 308 mA cm$^{-2}$, which exceeds the United States Department of Energy (DOE) target[48]. Meanwhile, the hydrogen-air fuel cell works stably at 0.6 V for 24 h (Fig. 5e). Compared with previously reported carbon-supported ORR catalysts (Fig. 5f), this PtCo@CoNC/NTG also exhibits a top-level performance in both current intensity and power density at a similar or even lower Pt loading (Supplementary Table 4).

## Theoretical calculation and mechanism analysis

The inherent advantages and composition differences of the PtCo@CoNC and Pt/C catalysts are analyzed (Fig. 6a). Obviously, the integrated construction of Pt-Co alloy and graphitic carbon would alleviate the accumulation, detachment and dissolution of metal nanoparticles compared with commercial carbon-supported catalysts. More importantly, PtCo@CoNC hybrid is significantly superior to commercial Pt/C with independent active sites in terms of active site richness and intrinsic properties, as multiple active sites, including Pt-Co, Co-N and C-N, synergistically promote the ORR process[49]. We then employ density functional theory (DFT) calculations to understand the ORR activity of PtCo@CoNC/NTG (Supplementary Fig. 21)[50]. The determined potential energy profiles

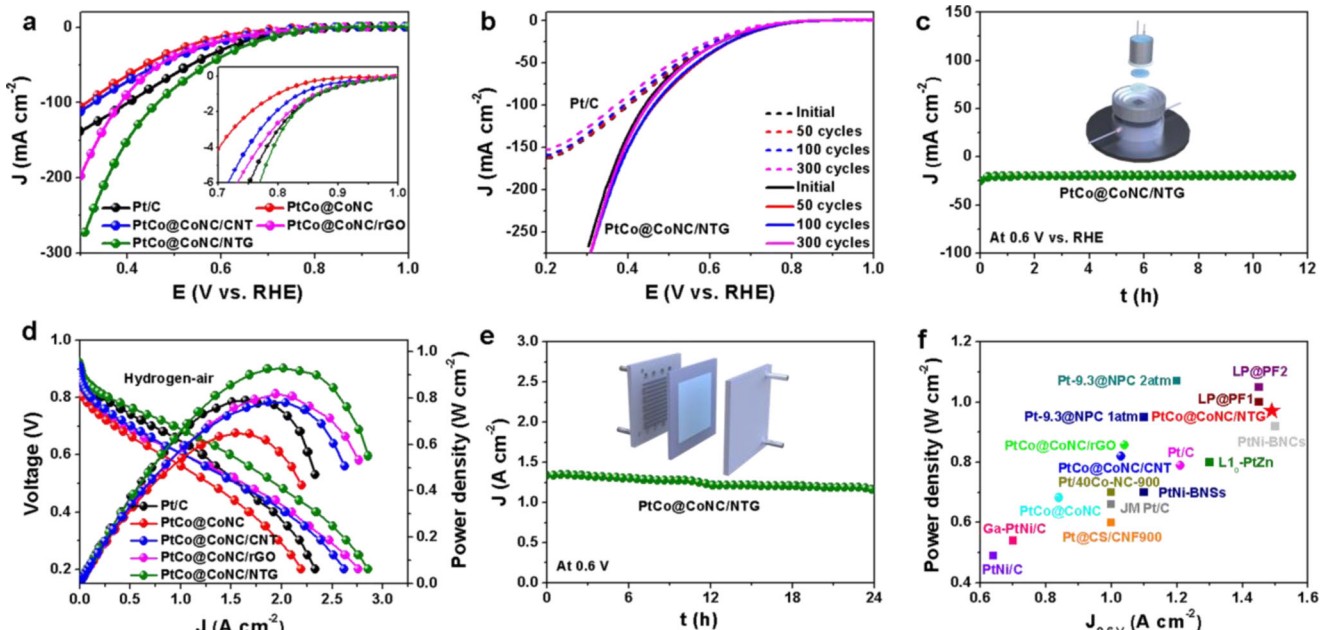

**Fig. 5 | Hydrogen-air fuel cell performance evaluation. a** LSV curves of different catalysts; inset a is an enlarged view of 0.7–1.0 V, **b** LSV curves of Pt/C and PtCo@CoNC/NTG before and after ADTs, and **c** I-t curve of PtCo@CoNC/NTG at 0.6 V vs. RHE in the GDE evaluation. **d** Fuel cell polarization plots of different cathode catalysts, **e** fuel cell stability test at a voltage of 0.6 V, and **f** performance comparison of recent carbon supported electrocatalysts in hydrogen-air fuel cells (references corresponding to data points are listed in Supplementary Table 4).

show that the Pt-Co (0.38 eV) interface possesses a lower overpotential than Pt (0.82 eV), which proves the better catalytic performance of alloyed Pt-Co (Fig. 6b). Additionally, the Co-N-C sites deliver evident ORR catalytic activity (Supplementary Fig. 22)[51]. Thus, we consider the synergistic role of Pt-Co and Co-N-C sites in the ORR process, especially some possible $H_2O_2$ intermediates that would be generated through a 2e pathway at Co-N-C sites (Supplementary Fig. 23)[52]. According to the DFT analysis, these $H_2O_2$ intermediates generated at the Co-N-C site are preferentially released and migrate to the adjacent Pt-Co sites for the subsequent 2e reaction, thereby achieving complete oxygen reduction (Fig. 6c, d). It is worth noting that, with the joint efforts of Pt-Co and Co-N-C sites, in addition to the traditional 4e pathway, the migration of the adsorbed products at the active sites realizes a synergistic 4e reaction[53], which provides more options for the ORR pathway and accelerated catalytic efficiency.

## Discussion

In summary, we report the multiscale design and integration engineering of an efficient platinum-nanocarbon hybrid electrocatalyst for hydrogen-air fuel cells. This platinum-carbon integrated catalyst demonstrates a high mass activity of 1.52 A $mg_{Pt}^{-1}$, which is 11.7-fold superior to commercial Pt/C (0.13 A $mg_{Pt}^{-1}$). More importantly, this robust integrated electrocatalyst shows remarkable durability, with only 1.3% decay after 30,000 potential cycles. The hydrogen-air fuel cell assembled by the PtCo@CoNC/NTG cathode achieves a current density of 1.50 A $cm^{-2}$ at 0.6 V and a power density of 980 mW $cm^{-2}$. Advanced analyses, including AC-STEM, XAFS, DFT calculations and electrochemical evaluations, clarify that the integrated catalysts containing multidimensional architectures and polyatomic active sites collectively strengthen the synergistic ORR catalytic activity and stability. Moreover, the improved mass transport brought by the unblocked transmission channels enables the efficient operation of integrated catalysts in hydrogen-air fuel cells. This work may provide valuable insights into the design and fabrication of high-performance Pt integrated catalysts for fuel cells and other energy technologies.

## Methods

### Reagents

Platinum acetylacetonate [Pt(acac)$_2$, 97%], cobalt nitrate hexahydrate [Co(NO$_3$)$_3$·6H$_2$O, 98%], 2-methylimidazole (C$_4$H$_6$N$_2$, 99%), and triethylamine (TEA, 99%) were purchased from Aladdin and were not further purified in the experiments. The solvents used in the experiment, such as triethylamine, perchloric acid, methanol, ethanol, and Nafion (5%, DuPont), were analytical reagents (ARs) and used directly without further purification. The deionized water used in the electrochemical measurements was ultrapure (18.2 MΩ).

### Preparation of PtCo@CoNC/NTG

Pt(acac)$_2$ (20 mg), NaCl (0.5 g) and rGO (10 mg) were dissolved in a mixed solution of 5 mL of H$_2$O and 5 mL of ethanol. The mixture was thoroughly stirred, dried and annealed at 260 °C for 2 h in an air atmosphere. The salt template was washed with a mixture of water and methanol 3 times and finally dried to obtain highly dispersed Pt/rGO. Then, 20 mL of Co(NO$_3$)$_2$·6H$_2$O (873 mg) methanol solution, 20 mL of 2-methylimidazole (985 mg) methanol solution and 10 μL triethylamine were separately put into 10 mL methanol containing Pt/rGO, followed by stirring at room temperature for 2 h. Pt@ZIF-67/rGO was obtained by centrifugation and washed with methanol 3 times. Subsequently, Pt@ZIF-67/rGO undergoes pyrolysis at 800 °C for 3 h under a 5% H$_2$/Ar atmosphere to collect PtCo@CoNC/NTG. Finally, PtCo@CoNC/NTG was achieved by 0.5 M sulfuric acid treatment overnight. Similarly, the synthesis of PtCo@CoNC/rGO only changed the amount of Co(NO$_3$)$_2$·6H$_2$O (218 mg) and 2-methylimidazole (246 mg). The synthesis of PtCo@CoNC/CNT turns rGO into CNT, and the preparation of PtCo@CoNC is achieved by ZIF-67 adsorption of Pt. The other synthesis and postprocessing steps were the same as those in the synthesis of PtCo@CoNC/NTG.

### Physical characterization

SEM was performed on an SU8010 (Hitachi, Japan) instrument at an accelerating voltage of 5 kV. TEM was conducted on a Tecnai G2-F30 instrument (FEI, Netherlands) with a field emission gun operating at

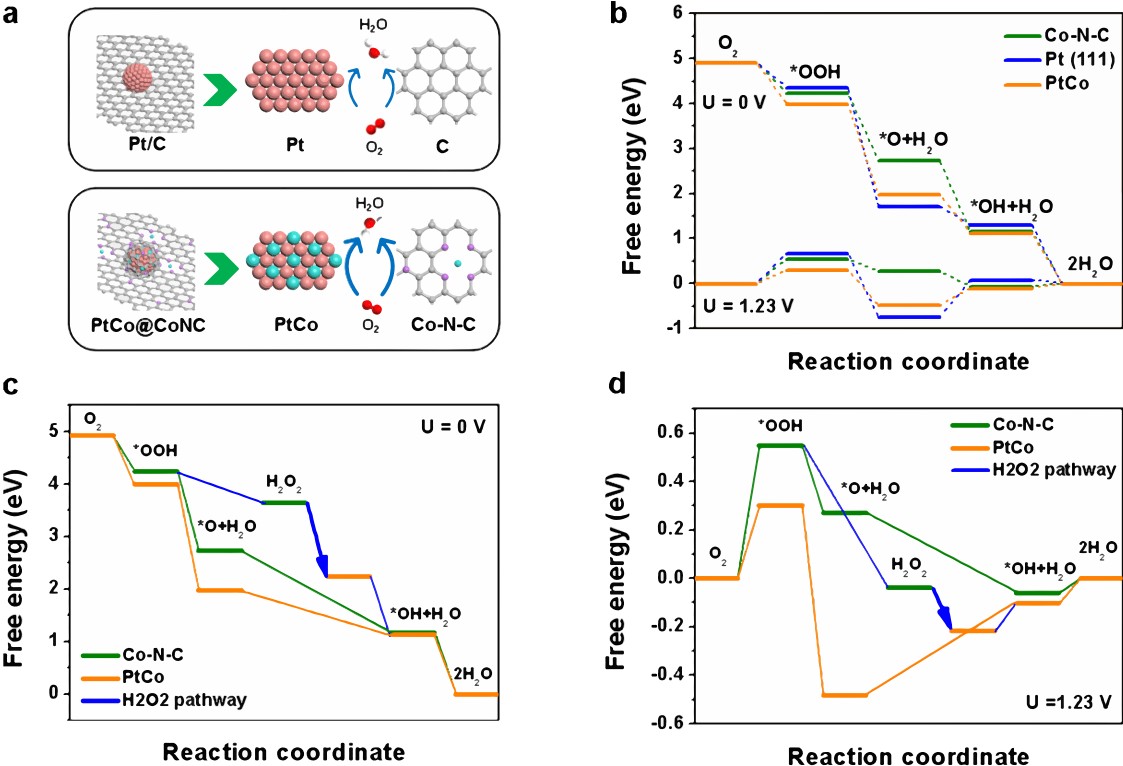

**Fig. 6 | Theoretical calculations. a** Schematic diagram of Pt/C and PtCo@CoNC for the synergistic ORR. **b** Free-energy diagram of the ORR pathways for Co-N-C, Pt(111) and PtCo sites. **c**, **d** Synergistic ORR catalytic pathways over Co-N-C and PtCo sites at 0 (**c**) and 1.23 V (**d**).

300 kV for TEM, HRTEM, HAADF-STEM and EDS patterns. For atomic-resolution imaging and EELS, the measurements were carried out on an AC-HAADF-STEM equipped with JEOL dual 158 mm² SDD detectors (JEOL Grand ARM300). XRD measurements were conducted in a SmartLab-SE X-ray diffractometer (Rigaku, Japan) with Cu Kα radiation ($\lambda = 1.5405$ Å) at 40 kV and 30 mA and a sweep speed of 10° min⁻¹. XPS measurements were collected with an ESCALAB 250 Xi (Thermo Fisher Scientific, USA) and a monochromatic Al X-ray source to analyze the chemical composition of the samples, and the test result was corrected according to the binding energy of C 1s (284.8 eV). Synchrotron X-ray spectroscopy was performed at the National Synchrotron Radiation Research Center (NSRRC), Taiwan. XANES and EXAFS at the Pt L₃-edge and Co K-edge were recorded with fluorescence mode at TLS-17C, where the energy resolutions of Pt and Co are 0.55 and 0.50 eV, respectively. The Brunauer–Emmett–Teller (BET, Micromeritics ASAP 2020, USA) test system was used to analyze the nitrogen adsorption-desorption isotherms, specific surface area and pore width distribution of the samples. Raman spectra were recorded on a LabRAM HR800 instrument (HORIBA Jobin Yvon, France) with a laser source of 532 nm, in which the ratio of $I_D/I_G$ was obtained based on the integral area. The Pt loadings were determined by inductively coupled plasma–mass spectrometry (ICP–MS, Agilent 7700, Agilent Technologies).

### RDE evaluation

An electrochemical workstation (Autolab PGSTAT302N) with an RDE was used to evaluate the electrocatalytic activity in a three-electrode electrochemical test system, which consisted of a glassy carbon electrode (0.196 cm²) covered with a catalyst film (working electrode), a Pt plate (counter electrode), and a RHE (reference electrode). To prepare the working electrode, 5 mg of catalyst was evenly dispersed in a mixture solution containing 0.9 mL of ethanol and 100 μL of 0.5 wt% Nafion solution to form a homogeneous catalyst ink, followed by 10 μL of catalyst ink pipetted onto the

clean glassy carbon surface until a thin film was formed. The electrochemical measurements were conducted in 0.1 M HClO₄ solution for the ORR at 25 °C. CV curves were collected from 0.05 to 1.1 V in the Ar-saturated 0.1 M HClO₄ solution at a scan rate of 50 mV s⁻¹. LSV curves were recorded in O₂-saturated 0.1 M HClO₄ solution at 1,600 rpm with a sweep rate of 10 mV s⁻¹. Significantly, the calculation of mass activity was based on the Pt loadings on the disk and the kinetic current at 0.9 V vs. RHE. The ADT was conducted in O₂-saturated 0.1 M HClO₄ solution by performing cyclic potential sweeps between 0.6 and 1.1 V at a sweep rate of 100 mV s⁻¹ for 30,000 cycles. The solution resistance was measured at open circuit voltage, and the final ORR polarization curve was obtained by manual iR correction. In addition, rotating ring disk electrode (RRDE) is used to detect the electron transfer number and the H₂O₂ yield during the ORR, and the collection efficiency of the ring is 0.37.

### GDE evaluation

An electrochemical workstation (Autolab PGSTAT302N) was used to evaluate the electrocatalytic activity in a three-electrode electrochemical test system, which consisted of a carbon paper electrode covered with a catalyst film (0.502 cm², working electrode), a Pt filament (counter electrode), and an Ag/AgCl electrode (reference electrode). Similarly, the catalyst ink pipetted onto the carbon paper surface to form a homogeneous working electrode, and Pt loadings were controlled at 0.02 mg cm⁻². The electrochemical measurements were conducted in 4.0 M HClO₄ solution for the ORR. CV activation is performed from 0.05 to 1.1 V vs. RHE in the Ar-saturated 4.0 M HClO₄ solution at a scan rate of 100 mV s⁻¹. LSV curves were recorded in O₂-saturated 4.0 M HClO₄ solution with a sweep rate of 50 mV s⁻¹. The ADT was conducted in O₂-saturated 4.0 M HClO₄ solution by performing cyclic potential sweeps between 0.05 and 1.1 V vs. RHE at a sweep rate of 100 mV s⁻¹ for 300 cycles. Throughout the GDE evaluation, the room temperature was maintained at 25 °C.

## MEA evaluation

The single hydrogen-air cell performance was detected in an MEA testing system (Arbin Instruments). The catalyst ink was prepared by ultrasonically mixing the catalyst, isopropanol and 5 wt% Nafion solution for 1 h, and the alcohol-water ratio was controlled to be 3:1, followed by spraying the ink on one side of a pretreated DF260 Nafion membrane (15 μm, Dongyue) to form the cathode catalyst layer of 4.0 cm$^2$. A commercial Pt/C (JM 40 wt%) catalyst was prepared by a similar method and sprayed on the other side of the membrane as the anode. The Pt loading at the anode was 0.1 mg cm$^{-2}$, and the Pt loading at the cathode was confirmed to be 0.1 mg cm$^{-2}$. Subsequently, commercial carbon paper (SGL-29BC) was directly used as the gas diffusion layer (GDL). The single cell was assembled with the prepared MEAs and two pieces of GDLs by using 600 lbs of hot pressing at 120 °C for 5 min. Pure hydrogen (99.999%, 400 mL min$^{-1}$) and compressed air (600 mL min$^{-1}$) were employed for the anode and cathode for the hydrogen-air fuel cell test. The single cell was preactivated at 0.5 V for 3 h before the polarization curve test, until the current density reached a steady state. The cell operating temperature throughout the MEA test was maintained at 70 °C, the relative humidity reached 100%, and the back pressure of both the anode and cathode was 0.2 MPa. Significantly, the hydrogen-air fuel cell measurements were not iR corrected, and the power density was equal to the current density times the voltage.

## Data availability

The data that support the findings of this study are available from the corresponding author upon reasonable request.

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

## Acknowledgements

This work is funded by the National Key Research and Development Program of China (2021YFA1501000, 2021YFA1600800), the National Natural Science Foundation of China (22075092), the Program for HUST Academic Frontier Youth Team (2018QYTD15) and the Innovation and Talent Recruitment Base of New Energy Chemistry and Device (B21003). We also acknowledge the support of the Analytical and Testing Center of HUST for XRD, SEM, TEM and XPS measurements and the NSRRC for synchrotron X-ray spectroscopies.

## Author contributions

B.Y.X. conceived the proposal, designed the experiments and supervised the project. L.H. carried out the sample synthesis, characterization and measurements. M.W. performed DFT calculations. C.-L.D and C.-C.Y. provided XAS analysis. D.D. and C.C. performed the fuel cell measurements. R.Q., M.W., C.X., B.Y., S.Z., F.-M.L. participated in the physical characterization and discussion of the results. B.Y.X. and L.H. wrote and revised the manuscript. All the authors contributed to the whole manuscript.

## Competing interests

The authors declare no competing interests.
