## [Peer Review File · Nature Communications]

An Integrated Platinum-Nanocarbon Electrocatalyst for Efficient Oxygen ReductionREVIEWER COMMENTS

Reviewer #1 (Remarks to the Author):

This paper demonstrated the integrated platinum-nanocarbon electrocatalyst for ORR with high efficiency. Authors use XAS to confirm the co-existence of PtCo alloy and atomic dispersed Co-N-C structure in their materials. However, there are some information missing, making the evidence weak to support the existence of Co-N-C structure and PtCo alloy formation. Overall, I believe it could be published in Nature Communications after major revision. Please find the following detailed comments:

1. Please be careful about the physical characterization in supporting information: which metal edge was measured? Which experiment was used? The details for XAS data analysis including EXAFS fitting and Wavelet transfer are missing.
2. Please include the EXAFS fitting results for k-space. This is critical to judge the goodness of fitting.
3. Please include all EXAFS fitting errors, including errors for bonding length, debye-waller factor, and energy shift.
4. For the fitting results of Pt L-edge and Co K-edge for PtCo@Co-N-C, the debye-waller factor for Pt-Co scattering and Co-N scattering is very small, which is even smaller than the limit 0.0008. It would cause some unrealistic fitting results in the coordination number or other parameters.
5. The fitting results demonstrate that it would have 4 coordinated Co-N and 8 coordinated Co-Co, but it is hard to see any contribution from Co-N. Also, the Wavelet transfer analysis cannot confirm the existence of Co-N. The difference in XANES also may be caused by the PtCo alloy formation instead of Co-N-C structure. The current results cannot support the existence of Co-N-C structure.
6. We could observe the Pt-Co scattering based on Pt L-edge fitting, but such scattering is missing in the Co K-edge fitting.
7. The shorter bonds in the PtCo@Co-N-C may indeed stand for the formation of PtCo alloy, but the wavelet transfer only shows one broad beak around 2Å. The 2D wavelet transfer is used to distinguish the Pt-Co and Pt-Pt bonding with high k-resolution. In addition, the wavelet transfer can be used to separated Co-Pt, Co-Co, and Co-N bonding with either high r-resolution or high k-resolution.

Reviewer #2 (Remarks to the Author):

I do not find the results as reported in this paper worth publishing in a high impact journal, at least not in the present form. The reasons are as follow:

To begin with, performance data need to be discussed within reasonable context. For example, both the halfwave potential and current density (Fig 4a, 4b) depend on the roughness factor of the electrode; therefore, one shall compare PGM mass or ECSA (H-UPD or CO-UPD) normalized data. This deficiency pertains to the performance discussions in GDE and MEA. Lacking mass loading or roughness factor values, the comparison of Pol curves does not help differentiating one catalyst system from another. Speaking of absolute values of reported performance, neither the mass activity in RDE or the current density in MEA are impressive; unless the RDE is done under sulfuric acid or the current density is achieved with a mass loading below 0.1 mg/cm².

Assuming PtCo particles are the same among samples, the present work aims at demonstrating the benefits of a 3D/multiscale catalyst/support design. To study the effect of various catalyst support structures regarding performance of oxygen reduction reaction, proof needs to be provided towards supporting the grand hypothesis that the catalyst, PtCo, is the same across samples. For example, TEM/XAS characterization must be given with respect to PtCo catalysts in the context of 0D/1D/2D support to confirm reasonably close dimensions and chemical components.

The theory discussion is out of focus. Figure 6a is not elaborated in the text but is quite misleading with respect to analogy of child/mother to bare carbon (little ORR activity)/Co-N-C (an ORR catalyst).

The authors demonstrated some reaction pathway calculations and some finite element multi-physics. However, how these simulations are quantitatively (or at least qualitatively) related to the benefits of the 3D structures is less than clear.

Reviewer #3 (Remarks to the Author):

In this article, the authors reported a preparation of synergistic catalyst using the combination of Pt deposited over graphene oxide and Co-ZIF to promote nanotube growth while adding Co-N-C PGN-free active sites. They carried out extensive studies including catalyst preparation, structural characterization and computational modeling. It is this reviewer's opinion that the article can be published by Nature Communication, only after the satisfactory revisions of some major deficiencies, as outlined in the following:

1. It is this reviewer's opinion that the definitions of 3-D, 2-D, 1-D and 0-D catalysts included in this article are somewhat arbitrary and do not actually reflect the catalyst morphologies. There is no clear morphological difference between them, at least between 2-D and 3-D, other than carbon nanotube was observed over the 3-D catalyst. Such growth of CNT in the presence of transition metal Co and organic moiety is well-known. Authors are recommended to find better and more rigorous definitions/descriptions for these catalysts instead what are used in the current manuscript.
2. There are some important experimental parameters missing. Key parameters, for example, include the platinum loading in the "as synthesized" catalysts and the Pt loading on the disk of the RDE used in the study (although the loadings were given for GDE and MEA tests). Without such info, it is difficult for readers to examine closely the data and to duplicate the study.
3. For the RDE measurement, at what voltage the mass activity was measured and over what Pt loading on the disk?
4. Between Line 199 to 201, several statements need to be corrected. DOE target for Pt mass activity in MEA should be measured at 0.9V iR-free, instead of 0.85 V. The current density at 0.8V should be measured directly using the actual cell voltage instead of iR corrected voltage. Also, when use DOE target, correct link to the webpage should be cited in the reference section.
5. Similarly, the pressure used in DOE MEA/fuel cell measurement protocol with either oxygen or air as oxidant should be at 1 bar absolute, or 0.5 bar for the back pressure after including 0.5 bar water pressure at 80 C. Authors used 0.2 MP or 2 bars as the back pressure, which indicate that the actual oxygen or air pressure is 2.5 bar, significantly higher than that defined by DOE test protocol. It is okay in my opinion for Authors to use such high pressure. However, Authors should not consider that MEA performance is meeting DOE target since it was not measured under the correct pressure. Higher pressure leading to a better fuel cell polarization is a well-known fact. To this end, Authors should also label the pressure they actually used in the supplementary table when making comparison with others.
6. PtCo alloy and Co-N-C synergetic catalysis was first proposed by Chong et. al. (Ref. 47, Science 2018) over almost the exact catalyst system presented here by Authors. In fact, many structural properties reported in this manuscript are very similar to that reported by Chong, et. al. Ref 44 was about synergistic integration of phosphonated poly(pentafluorostyrene) into membrane. It has absolutely nothing to do with the Pt/PGM-free catalyst interaction and should not be used as the refernece. Proper reference and credit should be given in the revised version.

It is this reviewer's opinion that these deficiencies should be corrected in the revised manuscript before it can be re-evaluated for the publication at Nature Comm.

Response to Reviewers

We really appreciate the the reviewers very much for your valuable time and insightful comments, which are extremely helpful in improving the quality of our manuscript and enhancing our future work. We also hope Reviewers will agree with our efforts, and the revised manuscript will be considered for publication in *Nature Communications*. Thank you very much.

Reviewer comments

Reviewer #1 (Remarks to the Author):

This paper demonstrated the integrated platinum-nanocarbon electrocatalyst for ORR with high efficiency. Authors use XAS to confirm the co-existence of PtCo alloy and atomic dispersed Co-N-C structure in their materials. However, there are some information missing, making the evidence weak to support the existence of Co-N-C structure and PtCo alloy formation. Overall, I believe it could be published in Nature Communications after major revision. Please find the following detailed comments:

Response: Thank you for your professional opinions on XAS characterization. We added complete information from experiments and analysis, updated almost all XAS data and fittings, and corrected some inappropriate descriptions in the manuscript to support our conclusions. We hope these corrections and our response will satisfy you, and thank you again for your constructive comments.

1. Please be careful about the physical characterization in supporting information: which metal edge was measured? Which experiment was used? The details for XAS data analysis including EXAFS fitting and Wavelet transfer are missing.

Response: Thanks for your kind comments. We provided more complete XAS characterization information, and other physical characterizations have also been rechecked. Notably, we have supplemented a separate section of XAS data analysis for XANES correction, EXAFS fitting and Wavelet transform in revised Supplementary information.

In the revised part: Page S3-S4, Supplementary information

The synchrotron X-ray spectroscopies were performed at the National Synchrotron Radiation Research Center (NSRRC), Taiwan. The X-ray absorption near edge structure (XANES) and the extended X-ray absorption fine structure (EXAFS) at Pt L₃-edge and Co K-edge were recorded with fluorescence mode at TLS-17C,

where the energy resolutions of Pt and Co are 0.55 and 0.50 eV, respectively.

XAS data analysis: Pt foil, PtO₂, Co foil and CoO were used as standard samples to evaluate the coordination environment and chemical state of Pt L₃-edge and Co K-edge of PtCo@CoNC samples. According to the energy correction of Pt foil and Co foil, the signals of Pt L₃-edge and Co K-edge were optimized using the Athena program in IFEFFIT software package, and then E₀ was determined based on the highest value of the first derivative of XANES. Through the corrected XANES white line peak position, the chemical valence range of Pt and Co in PtCo@CoNC/NTG could be determined. For EXAFS analysis, the k space range (2.0-12.0 Å⁻¹) was selected for Fourier transform (FT) to obtain R space, where the weight of k was 3. The first shell of R space was selected for inverse Fourier transform to obtain q space, involving Pt L₃-edge of Pt foil (1.1-3.1 Å), PtCo@CoNC/NTG (1.0-3.0 Å), PtO₂ (1.1-2.1 Å), and Co K-edge of Co foil (1.0-3.0 Å), PtCo@CoNC/NTG (1.0-3.3 Å) and CoO (1.1-3.0 Å). The continuous Cauchy wavelet transform (WT) analysis (R range = 0.2-6.0 Å; k range = 0-13.0 Å⁻¹) of EXAFS spectra was based on k³-weighted k space.¹ By narrowing the selection range of k and R space, high-resolution WT analysis of PtCo@CoNC/NTG could be obtained.² In the subsequent EXAFS fitting, the value range of k and R space was the same as the selection in the above-mentioned FT and inverse Fourier transform. According to the set coordination number of Pt foil (12) and Co foil (12), the amplitude reduction factor S₀² of Pt L₃-edge and Co K-edge was obtained by fitting of Artemis program. After EXAFS fitting, the detailed fitting information of PtCo@CoNC/NTG could be recorded, including coordination number, bonding length, debye-waller factor, and energy shift. etc. More EXAFS fitting information are shown in Supplementary Table 1 and Table 2.

2. Please include the EXAFS fitting results for k-space. This is critical to judge the goodness of fitting.

Response: As you mentioned, the fitting of k-space is closely related to the coordination and bond length in subsequent R-space fitting. Noticeably, our R-space fitting mainly focuses on the first shell, so the fitting result of k-space is in error with the original data. We fitted the obtained k-space and marked the fitting range

of different samples, hoping to reach a consensus, thanks.

In the revised part: Page S21 and S23, Supplementary information

Supplementary Figure 14. k, R space and inverse FT-EXAFS fitting results of Pt L₃-edge for (a-c) Pt foil (FT range: 2.0-12.0 Å⁻¹; fitting range: 1.1-3.1 Å), (d-f) PtCo@CoNC/NTG (FT range: 2.0-12.0 Å⁻¹; fitting range: 1.0-3.0 Å), and (g-i) PtO₂ (FT range: 2.0-12.0 Å⁻¹; fitting range: 1.1-2.1 Å).

Supplementary Figure 16. k, R space and inverse FT-EXAFS fitting results of Co K-edge for (a-c) Co foil (FT range: 2.0-12.0 Å⁻¹; fitting range:1.0-3.0 Å), (d-e) PtCo@CoNC/NTG (FT range: 2.0-12.0 Å⁻¹; fitting range:1.0-3.3 Å), and (g-i) CoO (FT range: 2.0-12.0 Å⁻¹; fitting range:1.1-3.0 Å).

3. Please include all EXAFS fitting errors, including errors for bonding length, debye-waller factor, and energy shift.

Response: This is constructive for our EXAFS fitting analysis. We added the fitting errors of bonding length, debye-waller factor, and energy shift in Supplementary Table 1 and Table 2. Taking into account the other coordination situations you mentioned, the fitting results and parameters we give are completely updated.

In the revised part: Page S38-S39, Supplementary information

Supplementary Table 1. Structural parameters of Pt foil, PtCo@CoNC/NTG, and PtO₂ extracted from the EXAFS fitting ($S_0^2=0.77$).

Sample	Bond	CN	R/Å	ΔE_0 /eV	$\sigma^2/\text{Å}^2$	R-factor
Pt foil	Pt-Pt	12*	2.763 ± 0.002	7.31 ± 0.49	0.0043 ± 0.0002	0.002
PtCo@CoNC/	Pt-Pt	4.91 ± 1.55	2.696 ± 0.006	5.65 ± 0.74	0.0063 ± 0.0015	0.001

NTG	Pt-Co	5.04 ± 0.58	2.578 ± 0.004		0.0051 ± 0.0006	
PtO ₂	Pt-O	5.17 ± 0.70	2.017 ± 0.008	12.51 ± 1.63	0.0011 ± 0.0010	0.010

Note: S_0^2 is the amplitude reduction factor (obtained by the fitting of Pt foil); CN is the coordination number; R is interatomic distance (the bond length between Pt central atoms and surrounding coordination atoms); ΔE_0 is energy shift; σ^2 is Debye-Waller factor (a measure of thermal and static disorder in absorber-scatterer distances); R-factor is used to value the goodness of the fitting; * the experimental EXAFS fit of metal foil by fixing CN as the known crystallographic value.

For Pt L₃-edge of PtCo@CoNC/NTG, data ranges: $2.0 < k < 12.0 \text{ \AA}^{-1}$, $1.0 < R < 3.0 \text{ \AA}$. The number of variable parameters is 7, out of total 12.44 independent points. The distance of Pt-Pt is based on the crystal structure of PtCo.

Supplementary Table 2. Structural parameters of Co foil, PtCo@CoNC/NTG, and CoO extracted from the EXAFS fitting ($S_0^2 = 0.85$).

Sample	Bond	CN	R/Å	ΔE_0 /eV	$\sigma^2/\text{Å}^2$	R-factor
Co foil	Co-Co	12*	2.492 ± 0.002	6.83 ± 0.40	0.0070 ± 0.0003	0.002
PtCo@CoNC/ NTG	Co-Co	4.45 ± 0.77	2.491 ± 0.013	6.40 ± 2.17	0.0052 ± 0.0013	0.010
	Co-N	1.53 ± 0.92	1.977 ± 0.063		0.0053 ± 0.0136	
	Co-Pt	0.33 ± 0.02	2.600 ± 0.050		-0.0185 ± 0.0553	
CoO	Co-O	4.48 ± 1.13	2.120 ± 0.013	0.17 ± 1.17	0.0073 ± 0.0028	0.010
	Co-Co	10.98 ± 1.53	3.018 ± 0.007		0.0079 ± 0.0010	

Note: S_0^2 is the amplitude reduction factor (obtained by the fitting of Co foil); CN is the coordination number; R is interatomic distance (the bond length between Co central atoms and surrounding coordination atoms); ΔE_0 is energy shift; σ^2 is Debye-Waller factor (a measure of thermal and static disorder in absorber-scatterer distances); R-factor is used to value the goodness of the fitting; * the experimental EXAFS fit of metal foil by fixing CN as the known crystallographic value.

For Co K-edge of PtCo@CoNC/NTG, data range: $2.0 < k < 12.0 \text{ Å}^{-1}$, $1.0 < R < 3.3 \text{ Å}$. The number of variable parameters is 10, out of total 14.38 independent points. The distances of Co-N and CoPt are based on the crystal structure of CoN and PtCo.

4. For the fitting results of Pt L-edge and Co K-edge for PtCo@Co-N-C, the debye-waller factor for Pt-Co scattering and Co-N scattering is very small, which is even smaller than the limit 0.0008. It would cause some unrealistic fitting results in the coordination number or other parameters.

Response: Thanks for your professional suggestions. We carefully rechecked the previous fitting results of PtCo@Co-N-C, paying special attention to the debye-waller factor. We hope the updated fitting results are satisfactory.

In the revised part: Page S38-S39, Supplementary information

Supplementary Table 1. Structural parameters of Pt foil, PtCo@CoNC/NTG, and PtO₂ extracted from the EXAFS fitting ($S_0^2 = 0.77$).

Sample	Bond	CN	R/Å	ΔE_0 /eV	$\sigma^2/\text{Å}^2$	R-factor
Pt foil	Pt-Pt	12*	2.763 ± 0.002	7.31 ± 0.49	0.0043 ± 0.0002	0.002
PtCo@CoNC/ NTG	Pt-Pt	4.91 ± 1.55	2.696 ± 0.006	5.65 ± 0.74	0.0063 ± 0.0015	0.001
	Pt-Co	5.04 ± 0.58	2.578 ± 0.004		0.0051 ± 0.0006	
PtO ₂	Pt-O	5.17 ± 0.70	2.017 ± 0.008	12.51 ± 1.63	0.0011 ± 0.0010	0.010

Note: S_0^2 is the amplitude reduction factor (obtained by the fitting of Pt foil); CN is the coordination number; R is interatomic distance (the bond length between Pt central atoms and surrounding coordination atoms); ΔE_0 is energy shift; σ^2 is Debye-Waller factor (a measure of thermal and static disorder in absorber-scatterer distances); R-factor is used to value the goodness of the fitting; * the experimental EXAFS fit of metal foil by fixing CN as the known crystallographic value.

For Pt L₃-edge of PtCo@CoNC/NTG, data ranges: $2.0 < k < 12.0 \text{ Å}^{-1}$, $1.0 < R < 3.0 \text{ Å}$. The number of variable parameters is 7, out of total 12.44 independent points. The distance of Pt-Pt is based on the crystal structure of PtCo.

Supplementary Table 2. Structural parameters of Co foil, PtCo@CoNC/NTG, and CoO extracted from the EXAFS fitting ($S_0^2 = 0.85$).

Sample	Bond	CN	R/Å	ΔE_0 /eV	$\sigma^2/\text{Å}^2$	R-factor
Co foil	Co-Co	12*	2.492 ± 0.002	6.83 ± 0.40	0.0070 ± 0.0003	0.002
PtCo@CoNC/ NTG	Co-Co	4.45 ± 0.77	2.491 ± 0.013	6.40 ± 2.17	0.0052 ± 0.0013	0.010
	Co-N	1.53 ± 0.92	1.977 ± 0.063		0.0053 ± 0.0136	
	Co-Pt	0.33 ± 0.02	2.600 ± 0.050		-0.0185 ± 0.0553	
CoO	Co-O	4.48 ± 1.13	2.120 ± 0.013	0.17 ± 1.17	0.0073 ± 0.0028	0.010
	Co-Co	10.98 ± 1.53	3.018 ± 0.007		0.0079 ± 0.0010	

Note: S_0^2 is the amplitude reduction factor (obtained by the fitting of Co foil); CN is the coordination number; R is interatomic distance (the bond length between Co central atoms and surrounding coordination atoms); ΔE_0 is energy shift; σ^2 is Debye-Waller factor (a measure of thermal and static disorder in absorber-scatterer distances); R-factor is used to value the goodness of the fitting; * the experimental EXAFS fit of metal foil by fixing CN as the known crystallographic value.

For Co K-edge of PtCo@CoNC/NTG, data range: $2.0 < k < 12.0 \text{ Å}^{-1}$, $1.0 < R < 3.3 \text{ Å}$. The number of variable parameters is 10, out of total 14.38 independent points. The distances of Co-N and CoPt are based on the crystal structure of CoN and PtCo.

5. The fitting results demonstrate that it would have 4 coordinated Co-N and 8 coordinated Co-Co, but it is hard to see any contribution from Co-N. Also, the Wavelet transfer analysis cannot confirm the existence of Co-N. The difference in XANES also may be caused by the PtCo alloy formation instead of Co-N-C structure. The current results cannot support the existence of Co-N-C structure.

Response: We are thankful for your professional suggestions. As you said, the previous result of EXAFS is not convincing to support the existence of Co-N-C structure, which is attributed to the comparison of the peak intensity of weak Co-N and strong Co-Co. Hence, we have reconsidered the Co-Pt coordination in Co K-edge

in the revised version, so that all coordination conditions of PtCo@CoNC/NTG have been rearranged. Furthermore, the updated WT analysis shows three centers (Co-N, Co-Co, Co-Pt), of which a weaker Co-N center is shown at $\sim 1.5 \text{ \AA}$, and the contribution of the Co-N structure can also be observed in the Co-N scattering path. Undoubtedly, the difference in XANES may be caused by PtCo alloy and Co-N-C structure together, in which the PtCo alloy plays a more important role. More importantly, we also used a series of other physical characterizations to confirm the existence of the Co-N-C structure, including AC-STEM, EELS and XPS. We hope these revisions and responses can further support our conclusion.

In the revised part: Page 7, 8 and 18, Revised manuscript

Pt in PtCo@CoNC/NTG shows the metallic form as seen from the similar white line intensity ($\sim 11,568 \text{ eV}$) with Pt foil in X-ray absorption near-edge structure (XANES) spectra, which matches well with XPS results (Fig. 3d).³⁴ Similarly, the Fourier transformed extended X-ray absorption fine structure (FT-EXAFS) for PtCo@CoNC/NTG at Pt L_3 -edge shows only one scattering peak at $\sim 2.2 \text{ \AA}$, assigned to the contribution of Pt-Pt/Pt-Co bonds (Supplementary Fig.14), which is smaller than that for Pt foil (Fig. 3e). Due to heteroatomic interaction in the Pt-Co alloying, a shortened Pt-Pt distance is displayed compared to the standard Pt foil (Supplementary Table 1), which is perfectly consistent with the fitted Pt-Pt and Pt-Co signals (Supplementary Fig. 15a).³⁵ Wavelet transformed EXAFS (WT-EXAFS) can simultaneously present k and R space information and further support a compressive Pt-Pt distance in PtCo alloys (Supplementary Fig. 15b). Compared with the WT diagrams of Pt foil and PtO₂, PtCo@CoNC/NTG shows an optimized coordination environment (Supplementary Fig. 15c), where the prominent center of $\sim 7.0 \text{ \AA}^{-1}$ and the weak center of $\sim 6.0 \text{ \AA}^{-1}$ are attributed to the interaction of PtCo alloy (Fig. 3f). By contrast, the white line intensity of Co K-edge in PtCo@CoNC/NTG ($\sim 7,726 \text{ eV}$) is located between Co foil and CoO, suggesting a partial oxidation of Co (Fig. 3g). AC-STEM observations of PtCo nanoparticles and Co single atoms simultaneously existing in PtCo@CoNC/NTG suggest that the oxidized Co species are contributed by Co single atoms (Fig. 3h).³⁶

Moreover, FT-EXAFS and WT-EXAFS at Co K-edge show a main peak at ~ 2.2 Å and some shoulder peaks at ~ 1.6 and ~ 2.6 Å (Supplementary Fig. 16, 17 and Table 2), which reasonably fit with the scattering paths of Co-Co, Co-N, and Co-Pt, respectively (Fig. 3i). These structural information support that the Co-N-C moieties and PtCo alloy sites in PtCo@CoNC/NTG would simultaneously act as the ORR active sites.³⁷⁻³⁹ The XAS spectrum analysis also show that other PtCo@CoNC integrated catalysts possess similar chemical components containing PtCo alloy and Co-N-C configuration (Supplementary Fig. 18, 19). Therefore, the 3D carbon carrier network in the PtCo@CoNC/NTG hybrid is unique compared to other PtCo@CoNC samples, which is conducive to smooth mass transport and continuous electron transfer.

Fig. 3 | Chemical state and coordination environment of PtCo@CoNC/NTG. **d** Pt L₃-edge XANES, and **e** the Fourier transforms of EXAFS spectra. **f** WT analysis of Pt L₃-edge in PtCo@CoNC/NTG. **g** Co K-edge XANES, and **h** the Fourier transforms of EXAFS spectra. **i** WT analysis of Co K-edge in PtCo@CoNC/NTG.

AC-STEM image, EELS spectrum, XPS N 1s and C 1s signals of PtCo@CoNC/NTG.

6. We could observe the Pt-Co scattering based on Pt L-edge fitting, but such scattering is missing in the Co K-edge fitting.

Response: Thanks for your valuable comments. The fitting of Co-Pt scattering is important to determine the PtCo alloy form. We considered the Co-Pt scattering path in the Co K-edge fitting, but Co-Pt scattering can only reach the weak signal due to the close distance between Co-Pt and Co-Co scattering and the strong influence of Co-Co scattering. Also, the form of PtCo alloy can be provided by the XRD, XPS, TEM, EDS characterizations in the manuscript.

In the revised part: Page S22 and S24, Supplementary information

Supplementary Figure 15. (a) The applied scattering paths of Pt L₃-edge for PtCo@CoNC/NTG, (b) EXAFS oscillations of Pt L₃-edge (k-weight = 3). (c) WT analysis of Pt for Pt foil, PtCo@CoNC/NTG and PtO₂ (R range = 0.2-6.0 Å; k range = 0-13.0 Å⁻¹).

Supplementary Figure 17. (a) The applied scattering paths of Co K-edge for PtCo@CoNC/NTG, (b) EXAFS oscillations of Co K-edge (k-weight = 3). (c) WT analysis of Co for Co foil, PtCo@CoNC/NTG and CoO (R range = 0.2-6.0 Å; k range = 0-13.0 Å⁻¹).

7. The shorter bonds in the PtCo@Co-N-C may indeed stand for the formation of PtCo alloy, but the wavelet transfer only shows one broad peak around 2Å. The 2D wavelet transfer is used to distinguish the Pt-Co and Pt-Pt bonding with high k-resolution. In addition, the wavelet transfer can be used to separate Co-Pt, Co-Co, and Co-N bonding with either high r-resolution or high k-resolution.

Response: Thanks for your constructive comments. We provided the 2D wavelet transfer with high k-resolution and R-resolution. Now, these separated bonding centers for Pt L₃-edge (Pt-Co, Pt-Pt) and Co K-edge (Co-N, Co-Co, Co-Pt) can be observed clearly. Thank you again for this valuable comment.

In the revised part: Page 18, Revised manuscript

Fig. 3 | Chemical state and coordination environment of PtCo@CoNC/NTG. d Pt L₃-edge XANES, and **e** the Fourier transforms of EXAFS spectra. **f** WT analysis of Pt L₃-edge in PtCo@CoNC/NTG. **g** Co K-edge XANES, and **h** the Fourier transforms of EXAFS spectra. **i** WT analysis of Co K-edge in PtCo@CoNC/NTG.

Reviewer #2 (Remarks to the Author):

I do not find the results as reported in this paper worth publishing in a high impact journal, at least not in the present form. The reasons are as follow:

1. To begin with, performance data need to be discussed within reasonable context. For example, both the halfwave potential and current density (Fig 4a, 4b) depend on the roughness factor of the electrode; therefore, one shall compare PGM mass or ECSA (H-UPD or CO-UPD) normalized data. This deficiency pertains to the performance discussions in GDE and MEA. Lacking mass loading or roughness factor values, the comparison of Pol curves does not help differentiating one catalyst system from another. Speaking of absolute values of reported performance, neither the mass activity in RDE or the current density in MEA are impressive; unless the RDE is done under sulfuric acid or the current density is achieved with a mass loading below 0.1 mg/cm².

Response: Thanks for your professional comments. As you mentioned, the comparison of half-wave potential and current density only considers the roughness of the electrode is not convincing. We consider that the PtCo@CoNC catalysts have PtCo and Co-N-C as the main intrinsic active sites, and the results based on ECSA (H-UPD or CO-UPD) can only express the Pt effect and would ignore the efforts of carbon and Co-N-C sites (Science 362, 2018, 1276). Therefore, we normalized the Tafel plots and mass activity based on the Pt loading on the working electrode (Fig. 4).

Furthermore, we provided the Pt loading on the electrode in the electrochemical evaluations of RDE, GDE and MEA in Supplementary Table 3. To clarify the performance comparison of different catalysts, the power current of 0.9 V is used in the RDE evaluation, and the polarization current of 0.6 V is used in the GDE evaluation to normalize the Pt loading to compare the mass activity of the catalyst. In the hydrogen-air single cell, the power density is mainly evaluated by controlling the Pt loading on the electrode, which is different from the evaluation standard of the hydrogen-oxygen fuel cell.

In addition, you mentioned mass activity in RDE or current density in MEA. Compared with many reported Pt alloy catalysts, the designed PtCo@CoNC integrated catalysts cannot show significant advantages in RDE evaluation. However, the multi-scale catalysts we designed pay more attention to their performance in MEA, which is the main reason why we design Pt-C integrated catalysts and optimize local mass transport (Fig. 1). In the hydrogen-air fuel cell, the PtCo@CoNC/NTG integrated catalyst exhibits superior power density under a loading of $0.1 \text{ mg}_{\text{Pt}} \text{ cm}^{-2}$ (Fig. 5).

In the revised part: Page 8 and 19, Revised manuscript; Page S40, Supplementary information

Fig. 4 | Electrochemical evaluation of PtCo@CoNC/NTG. a LSV curves, **b** Tafel plots, **c** half-wave potential and mass activity at 0.9 V vs. RHE of different Pt catalyst. **d** CV profiles, **e** LSV curves, and **f** activity comparison before and after ADTs of Pt/C and PtCo@CoNC/NTG.

Fig. 1 | Schematic diagram of improvement mechanism for PtCo@CoNC/NTG.

Fig. 5 | Hydrogen-air fuel cell performance evaluation. **a** LSV curves of different catalysts, inset **a** is an enlarged view of 0.7~1.0 V, **b** LSV curves of Pt/C and PtCo@CoNC/NTG before and after ADTs, and **c** I-t curve of PtCo@CoNC/NTG at 0.6 V vs. RHE in the GDE evaluation. **d** Fuel cell polarization plots of different cathode catalysts, **e** fuel cell stability test at the voltage of 0.6 V, and **f** performance comparison of recent carbon supported electrocatalysts in hydrogen-air fuel cells.

Supplementary Table 3. The electrochemical evaluations of commercial Pt/C and different PtCo@CoNC catalysts.

Catalyst	Pt mass ratio (wt%)	RDE			GDE			MEA			
		Pt loading ($\mu\text{g cm}^{-2}$)	Half-wave potential (V)	Mass activity _{0.9} ($\text{A mg}_{\text{Pt}}^{-1}$)	Pt loading ($\text{mg}_{\text{Pt}} \text{cm}^{-2}$)	Current density _{0.6} (mA cm^{-2})	Mass activity _{0.6} ($\text{A mg}_{\text{Pt}}^{-1}$)	Anode loading ($\text{mg}_{\text{Pt}} \text{cm}^{-2}$)	Cathode loading ($\text{mg}_{\text{Pt}} \text{cm}^{-2}$)	Current density _{0.6} (A cm^{-2})	Power density (mW cm^{-2})
Pt/C	40.0	23.2	0.87	0.13	0.04	30.2	0.75	0.1	0.2	1.21	780
PtCo@CoNC	3.2	8.1	0.86	0.25	0.02	9.4	0.47	0.1	0.1	0.84	680
PtCo@CoNC/CNT	2.9	7.4	0.88	0.42	0.02	16.1	0.81	0.1	0.1	1.03	820
PtCo@CoNC/rGO	2.5	6.3	0.89	0.86	0.02	17.5	0.88	0.1	0.1	1.04	850
PtCo@CoNC/NTG	4.5	11.5	0.94	1.52	0.02	38.9	1.95	0.1	0.1	1.50	980

2. Assuming PtCo particles are the same among samples, the present work aims at demonstrating the benefits of a 3D/multiscale catalyst/support design. To study the effect of various catalyst support structures regarding performance of oxygen reduction reaction, proof needs to be provided towards supporting the grand hypothesis that the catalyst, PtCo, is the same across samples. For example, TEM/XAS characterization must be given with respect to PtCo catalysts in the context of 0D/1D/2D support to confirm reasonably close dimensions and chemical components.

Response: This is a constructive comment for our present work. We have supplemented the TEM/XAS characterization of PtCo@CoNC catalysts to determine the same across samples. Supplementary electron microscopy characterization includes SEM, TEM, HRTEM, AC-HAADF-STEM and EDS mapping. It is clear that the PtCo@CoNC samples we mentioned in the manuscript are composed of PtCo alloy and graphite carbon containing Co-N-C sites, among which the atomic ratios of PtCo alloys are close to 1/1, the main difference lies in the PtCo@CoNC microstructure constructed by different graphitic carbon networks. In addition, the XAS spectrum analysis of other PtCo@CoNC samples shows that the multiscale designed PtCo@CoNC integrated catalysts possess similar chemical components containing PtCo alloy and Co-N-C configuration. Hence, the 3D carbon carrier network in the PtCo@CoNC/NTG hybrid is unique compared to other PtCo@CoNC samples.

In the revised part: Page S10-S15; S25-S26, Supplementary information

Supplementary Figure 3. The models and corresponding SEM images of (a) PtCo@CoNC, (b)

PtCo@CoNC/CNT, (c) PtCo@CoNC/rGO and (d) PtCo@CoNC/NTG.

Supplementary Figure 8. STEM images and corresponding EDS mapping of (a) PtCo@CoNC, (b) PtCo@CoNC/CNT, (c) PtCo@CoNC/rGO and (d) PtCo@CoNC/NTG.

Supplementary Figure 7. TEM, HRTEM and AC-STEM images of (a) PtCo@CoNC, (b) PtCo@CoNC/CNT, (c) PtCo@CoNC/rGO and (d) PtCo@CoNC/NTG.

Supplementary Figure 18. (a) Pt L₃-edge XANES, (b) k space, (c) R space, and (d) inverse FT-EXAFS for PtCo@CoNC hybrids (k-weight = 3; k range = 2.0-12.0 \AA^{-1}).

Supplementary Figure 19. (a) Co K-edge XANES, (b) k space, (c) R space, and (d) inverse FT-EXAFS for PtCo@CoNC hybrids (k-weight = 3; k range = 2.0-12.0 \AA^{-1}).

3. The theory discussion is out of focus. Figure 6a is not elaborated in the text but is quite misleading with respect to analogy of child/mother to bare carbon (little ORR activity)/Co-N-C (an ORR catalyst). The authors demonstrated some reaction pathway calculations and some finite element multi-physics. However, how these simulations are quantitatively (or at least qualitatively) related to the benefits of the 3D structures is less than clear.

Response: Thanks for your constructive comments. We revised the theory discussion that pays more attention to the synergistic ORR mechanism of PtCo and Co-N-C. Through DFT theoretical calculations, we mainly analyzed the adsorption and Gibbs free energy of the intermediate products in the intrinsic active sites of PtCo and Co-N-C. Compared with commercial carbon-supported Pt catalysts, PtCo@CoNC integrated catalysts have more advantages in active site density and synergistic ORR pathways.

According to your kind suggestions, we revised Fig. 6a and re-described the advantages of PtCo@CoNC integrated catalyst over commercial Pt/C catalyst in the revised manuscript. In addition, we modified the description of DFT calculation and physics simulation. Significantly, the DFT calculation mainly focuses on the catalytic efficiency and synergistic mechanism of the intrinsic catalytic activity of the PtCo@CoNC/NTG catalyst by qualitative analysis, which is helpful to understand the enhanced ORR performance. The benefits of the 3D structure are mainly the efficient utilization, continuous rapid electron transfer and locally optimized mass transport of active sites brought by the open space structure, continuous carbon network and nanotube channels (Fig. 1), which is helpful to improve ORR performance and MEA efficiency. The finite element simulation mainly shows that the porous channel design of the MEA cathode is important to enhance the transportation of oxygen and water and other substances, so as to qualitatively clarify the design advantages of our 3D structured catalyst with nanochannels. Thank you again for this valuable advice.

In the revised part: Page 10, 11 and 20, Revised manuscript

The inherent advantages and composition differences of PtCo@CoNC and Pt/C catalyst are analyzed (Fig.

6a). Obviously, the integrated construction of PtCo alloy and graphitic carbon would alleviate the accumulation, detachment and dissolution of metal nanoparticles compared with commercial carbon-supported catalysts. More importantly, the PtCo@CoNC hybrid is significantly superior to commercial Pt/C with independent active sites in terms of active sites richness and intrinsic properties, as the multiple active sites including Pt-Co, Co-N and C-N would promote the ORR process synergistically.

The determined potential energy profiles show that the Pt-Co (0.38 eV) interface possesses a lower overpotential than Pt (0.82 eV), which proves a better catalytic performance of alloyed Pt-Co (Fig. 6b). Also, the Co-N-C sites deliver evident ORR catalytic activity (Supplementary Fig. 26).

According to the DFT analysis, these H₂O₂ intermediates generated at the Co-N-C site are preferentially released and migrate to the adjacent Pt-Co sites for the subsequent 2e reaction, and achieve complete oxygen reduction (Fig. 6c, 6d).

When the single cell is operated at 0.6 V, the oxygen mole fraction varies negligibly along the thickness direction of the cathode, but it changes from 0.18 to 0 along the width of the electrode (Fig. 6e). In contrast, the distribution of water products on the cathode is from 0.15 to 0.4 (Fig. 6f). Accordingly, the flow field direction of oxygen and water is also consistent with the mass distribution (Supplementary Fig. 30).

Fig. 6 | Theoretical calculations and simulations. **a** Schematic diagram of Pt/C and PtCo@CoNC for synergistic ORR. **b** Free-energy diagram of the ORR pathways for Co-N-C, Pt(111) and PtCo sites. **c, d** Synergistic ORR catalytic pathways over Co-N-C and PtCo sites at 0 (c) and 1.23 V (d). **e-h** Finite element simulations of oxygen mole fraction (e), water mole fraction (f), pressure and velocity field (g) and current density (h).

Fig. 1 | Schematic diagram of improvement mechanism for PtCo@CoNC/NTG.

Reviewer #3 (Remarks to the Author):

In this article, the authors reported a preparation of synergistic catalyst using the combination of Pt deposited over graphene oxide and Co-ZIF to promote nanotube growth while adding Co-N-C PGN-free active sites. They carried out extensive studies including catalyst preparation, structural characterization and computational modeling. It is this reviewer's opinion that the article can be published by Nature Communication, only after the satisfactory revisions of some major deficiencies, as outlined in the following:

Response: Thanks for your interest in our work. Based on the major deficiencies you mentioned, including electrochemical evaluation, DOE target and proper references, we carefully correct them in the revision. We hope you will be satisfied with the revised manuscript, and thank you again for the professional comments.

1. It is this reviewer's opinion that the definitions of 3-D, 2-D, 1-D and 0-D catalysts included in this article are somewhat arbitrary and do not actually reflect the catalyst morphologies. There is no clear morphological difference between them, at least between 2-D and 3-D, other than carbon nanotube was observed over the 3-D catalyst. Such growth of CNT in the presence of transition metal Co and organic moiety is well-known. Authors are recommended to find better and more rigorous definitions/descriptions for these catalysts instead what are used in the current manuscript.

Response: This is a good comment. As you mentioned, the definitions of 3-D, 2-D, 1-D and 0-D catalysts do not accurately reflect the catalyst morphologies. Throughout the whole revised manuscript including description, Figures and Tables, we redefined these different PtCo catalysts as PtCo@CoNC/NTG, PtCo@CoNC/rGO, PtCo@CoNC/CNT and PtCo@CoNC, replacing the previous definition of 3-D, 2-D, 1-D and 0-D catalysts, respectively. We hope this updating can help readers to understand our catalyst design.

2. There are some important experimental parameters missing. Key parameters, for example, include the platinum loading in the "as synthesized" catalysts and the Pt loading on the disk of the RDE used in the study

(although the loadings were given for GDE and MEA tests). Without such info, it is difficult for readers to examine closely the data and to duplicate the study.

Response: This is a good comment to improve our work. The parameters you mentioned are crucial for evaluating ORR performance. We added relevant parameters in Supplementary Table 3.

In the revised part: S40, Supplementary information

Supplementary Table 3. The electrochemical evaluations of commercial Pt/C and different PtCo@CoNC catalysts.

Catalyst	Pt mass ratio (wt%)	RDE			GDE			MEA			
		Pt loading ($\mu\text{g cm}^{-2}$)	Half-wave potential (V)	Mass activity _{0.9} ($\text{A mg}_{\text{Pt}}^{-1}$)	Pt loading ($\text{mg}_{\text{Pt}} \text{cm}^{-2}$)	Current density _{0.6} (mA cm^{-2})	Mass activity _{0.6} ($\text{A mg}_{\text{Pt}}^{-1}$)	Anode loading ($\text{mg}_{\text{Pt}} \text{cm}^{-2}$)	Cathode loading ($\text{mg}_{\text{Pt}} \text{cm}^{-2}$)	Current density _{0.6} (A cm^{-2})	Power density (mW cm^{-2})
Pt/C	40.0	23.2	0.87	0.13	0.04	30.2	0.75	0.1	0.2	1.21	780
PtCo@CoNC	3.2	8.1	0.86	0.25	0.02	9.4	0.47	0.1	0.1	0.84	680
PtCo@CoNC/CNT	2.9	7.4	0.88	0.42	0.02	16.1	0.81	0.1	0.1	1.03	820
PtCo@CoNC/rGO	2.5	6.3	0.89	0.86	0.02	17.5	0.88	0.1	0.1	1.04	850
PtCo@CoNC/NTG	4.5	11.5	0.94	1.52	0.02	38.9	1.95	0.1	0.1	1.50	980

3. For the RDE measurement, at what voltage the mass activity was measured and over what Pt loading on the disk?

Response: Thanks for your professional comment. Voltage and Pt loading are important parameters for evaluating the mass activity of Pt-based electrocatalyst. For the RDE measurement, the mass activity of the catalyst is based on the kinetic current and Pt loading at 0.9 V vs. RHE. We supplemented more comprehensive RDE evaluation section in the supplementary electrochemical measurements section, and it is mentioned in the revised manuscript that the mass activity is based on 0.9 V vs. RHE. In addition, we marked the Pt loading of all Pt-based catalysts in the Supplementary Table 3. Thank you again for the valuable comments.

In the revised part: Page S4 and S40, Supplementary information

Linear sweep voltammetry (LSV) curves were recorded in the O₂-saturated 0.1 M HClO₄ solution at 1600 rpm with a sweep rate of 10 mV s⁻¹. Significantly, the calculation of mass activity was based on the Pt loadings on the disk and the kinetic current at 0.9 V vs. RHE. The accelerated durability test (ADT) was conducted in the O₂-saturated 0.1 M HClO₄ solution by performing cyclic potential sweeps between 0.6 and 1.1 V at a sweep rate of 100 mV s⁻¹ for 30,000 cycles.

Supplementary Table 3. The electrochemical evaluations of commercial Pt/C and different PtCo@CoNC catalysts.

Catalyst	Pt mass ratio (wt%)	RDE			GDE			MEA			
		Pt loading ($\mu\text{g cm}^{-2}$)	Half-wave potential (V)	Mass activity _{0.9} ($\text{A mg}_{\text{Pt}}^{-1}$)	Pt loading ($\text{mg}_{\text{Pt}} \text{cm}^{-2}$)	Current density _{0.6} (mA cm^{-2})	Mass activity _{0.6} ($\text{A mg}_{\text{Pt}}^{-1}$)	Anode loading ($\text{mg}_{\text{Pt}} \text{cm}^{-2}$)	Cathode loading ($\text{mg}_{\text{Pt}} \text{cm}^{-2}$)	Current density _{0.6} (A cm^{-2})	Power density (mW cm^{-2})
Pt/C	40.0	23.2	0.87	0.13	0.04	30.2	0.75	0.1	0.2	1.21	780
PtCo@CoNC	3.2	8.1	0.86	0.25	0.02	9.4	0.47	0.1	0.1	0.84	680
PtCo@CoNC/CNT	2.9	7.4	0.88	0.42	0.02	16.1	0.81	0.1	0.1	1.03	820
PtCo@CoNC/rGO	2.5	6.3	0.89	0.86	0.02	17.5	0.88	0.1	0.1	1.04	850
PtCo@CoNC/NTG	4.5	11.5	0.94	1.52	0.02	38.9	1.95	0.1	0.1	1.50	980

4. Between Line 199 to 201, several statements need to be corrected. DOE target for Pt mass activity in MEA should be measured at 0.9 V iR-free, instead of 0.85 V. The current density at 0.8V should be measured directly using the actual cell voltage instead of iR corrected voltage. Also, when use DOE target, correct link to the webpage should be cited in the reference section.

Response: We very much approve of your professional comments. As you said, the mass activity evaluation of DOE target is based on a voltage of 0.9 V, instead of 0.85 V. We removed mass activity comparisons and inappropriate description that are inconsistent with the DOE target protocol at the MEA level. In addition, we also quoted the DOE webpage as a reference, which is very necessary. Thank you again for your constructive comments.

In the revised part: Page 10 and 15, Revised manuscript

Noticeably, the polarization currents at 800 and 675 mV_{IR-free} are 308 and 1125 mA cm⁻², respectively, which exceeds the United States Department of Energy (DOE) target.⁴⁵ Meanwhile, the hydrogen-air fuel cell works stably at 0.6 V for 24 h (Fig. 5e).

45 Myers, D., Kariuki, N., Ahluwalia, R., Wang, X. & Peng, J. K. Rationally Designed Catalyst Layers for PEMFC Performance Optimization (US DOE, 2015); https://www.hydrogen.energy.gov/pdfs/review15/fc106_myers_2015_o.pdf

5. Similarly, the pressure used in DOE MEA/fuel cell measurement protocol with either oxygen or air as oxidant should be at 1 bar absolute, or 0.5 bar for the back pressure after including 0.5 bar water pressure at 80 C. Authors used 0.2 MP or 2 bars as the back pressure, which indicate that the actual oxygen or air pressure is 2.5 bar, significantly higher than that defined by DOE test protocol. It is okay in my opinion for Authors to use such high pressure. However, Authors should not consider that MEA performance is meeting DOE target since it was not measured under the correct pressure. Higher pressure leading to a better fuel cell polarization

is a well-known fact. To this end, Authors should also label the pressure they actually used in the supplementary table when making comparison with others.

Response: Thanks for your professional comment. The DOE MEA/fuel cell measurement protocol and performance standard are strict, our test needs to meet the corresponding test parameters to be comparable. As you mentioned, our hydrogen-air test protocol is significantly different from DOE, especially the pressure parameters. In the revised supplementary Table 4, we have marked the pressure parameters they actually used in the when making comparison with others.

In the revised part: Page S41, Supplementary Table 4, Supplementary information

Supplementary Table 4. Comparison of our PtCo@CoNC, commercial Pt/C and some carbon supported ORR electrocatalysts in hydrogen-air fuel cells.

Catalysts	Anode Pt loading (mg cm ⁻²)	Cathode Pt loading (mg cm ⁻²)	Current density at 0.6 V (mA cm ⁻²)	Power density (mW cm ⁻²)	Back pressure (MPa)	References
PtCo@CoNC/NTG	0.1	0.1	1.50	980		This work
PtCo@CoNC/rGO	0.1	0.1	1.04	850		
PtCo@CoNC/CNT	0.1	0.1	1.03	820	0.2	
PtCo@CoNC	0.1	0.1	0.84	680		
Pt/C	0.1	0.2	1.21	780		
JM Pt/C	0.1	0.3	1.0 #	664	H ₂ , 0.1; Air, 0.2	J. Am. Chem. Soc. 2017, 139, 17281
PtNi/C	0.15	0.15	0.64	490 #	without	Nano Lett. 2018, 18, 2450
Ga-PtNi/C	0.15	0.15	0.70	540 #		
PtNi-BNSs	0.1	0.15	1.0	700	0.2	Science 2019, 366, 850
PtNi-BNCs	0.1	0.15	1.5	920		
Pt/40Co-NC-900	N/A	0.13	1.1 #	700 #	H ₂ , N/A; Air, 0.1	Nano Lett. 2018, 18, 4163
L1 ₀ -PtZn	0.1	0.104	1.3 #	800 #	0.15	Adv. Energy Mater. 2020, 10, 2000179
Pt-9.3@NPC 1atm	0.1	0.1	1.1	947	0.1	
Pt-9.3@NPC 2atm	0.1	0.05	1.2	1071	0.2	ACS Energy Lett. 2020, 5, 3021
Pt/FeN ₄ -C	0.1	0.1	1.4	679		
Pt ₃ Co/FeN ₄ -C	0.1	0.1	1.6	822	0.15	Energy Environ. Sci. , 2021, 14, 4948
Coplanar Pt/C NMs	0.11	0.1	0.76 #	553	0.15	Angew. Chem. Int. Ed. 2021, 60, 6533
LP@PF1	0.35	0.033	1.45 #	1000		
LP@PF2	0.35	0.035	1.45 #	1100	0.2	Science 2018, 362, 1276

The data are not given, but excavated from the hydrogen-air fuel cell polarization curves

6. PtCo alloy and Co-N-C synergetic catalysis was first proposed by Chong et. al. (Ref. 47, Science 2018) over almost the exact catalyst system presented here by Authors. In fact, many structural properties reported in this manuscript are very similar to that reported by Chong, et. al. Ref 44 was about synergistic integration of phosphonated poly(pentafluorostyrene) into membrane. It has absolutely nothing to do with the Pt/PGM-free catalyst interaction and should not be used as the reference. Proper reference and credit should be given in the revised version.

Response: As you mentioned, the synergistic catalysis of PtCo alloy and Co-N-C is similar to that reported by Chong, et. al. Ref 44. Notably, our work is more focused on designing multi-scale open structures and transport channels at the microstructure level, and improving ORR efficiency by optimizing local mass transmission and diffusion (Fig. 1). In addition, we carefully checked the suitability of all references, removed some inappropriate (including Ref.44) and added some new references. Thank you again for your suggestions.

Fig. 1 | Schematic diagram of improvement mechanism for PtCo@CoNC/NTG.

In the revised part: Page 15, References, Revised manuscript

15 Zaman, S. et al. Oxygen Reduction Electrocatalysts toward Practical Fuel Cells: Progress and Perspectives. *Angew. Chem. Int. Ed.* **60**, 17832-17852 (2021).

20 Qiao, Z. et al. Atomically dispersed single iron sites for promoting Pt and Pt₃Co fuel cell catalysts:

performance and durability improvements. *Energy Environ. Sci.* **14**, 4948-4960 (2021).

- 46 Huang, L. et al. Boosting Oxygen Reduction via Integrated Construction and Synergistic Catalysis of Porous Platinum Alloy and Defective Graphitic Carbon. *Angew. Chem. Int. Ed.* **60**, 25530-25537 (2021).

REVIEWER COMMENTS

Reviewer #1 (Remarks to the Author):

I have carefully reviewed authors' response and believed my and other major concerns have been addressed. There is one minor thing that needs to be corrected before final publishing in Nature Communications.

In Supplementary Table S2, the σ^2 of Co-Pt bond in PtCo@CoNC/NTG is negative, -0.0185, which doesn't make sense. Please double check and reanalyze.

Reviewer #2 (Remarks to the Author):

The authors have made revisions on material characterizations and put more details on the electrochemical performance evaluation.

If this paper concludes that PtCo@CoNC/NTG is a potentially good catalyst, the reviewer agrees. The stability demonstrated by this non-PGM catalyst can be of interest to the fuel cell community; however, the performance on ORR activity and power density are not impressive, as shown by this new table added (Table S4). Given it is a PGM/non-PGM hybrid structure, there could be a potential question on whether the activity and stability performance is attributed to the PGM catalyst. 0.1 mg/cm² PGM loading is sufficient to deliver the activity and stability performance as shown by recent literature (for example Du et al. 10.1149/1945-7111/ac3598; Mauger et al. 10.1016/j.jpowsour.2021.230039).

If the paper is to study the role of non-PGM catalysts as catalyst support for PGM catalysts, the work lacks convincing evidence. Shown by Figure S7/S8, the PtCo particles have different sizes, is it possible that the authors saw a size-effect? If the authors fit the XAS on the Pt in Figure S18/S19, are there Pt-Co bond changes in CN and distances? Overall, the authors have not presented why PtCo@CoNC/rGO, PtCo@CoNC/CNT, and PtCo@CoNC are not as good as PtCo@CoNC/NTG with experimental evidence. As a result, the claims on the "multi-dimensional architectures and polyatomic active sites collectively strengthen the synergistic ORR catalytic activity and stability" appears groundless.

The reviewer cannot see the relevance of Fig 6e-h to the study, even after the revision. The scenario seems to be universal, not limited to a single catalyst system. The flow channels and the channels in the catalyst support are not at the same length scale, some details are required for helping the readers to see how the model applies.

Reviewer #3 (Remarks to the Author):

Authors made effort in their revised manuscript. I am generally fine with their responses except to my initial comment 4. My initial comment 4 is as the following:

"Between Line 199 to 201, several statements need to be corrected. DOE target for Pt mass activity in MEA should be measured at 0.9V iR-free, instead of 0.85 V. The current density at 0.8V should be measured directly using the actual cell voltage instead of iR corrected voltage. Also, when use DOE target, correct link to the webpage should be cited in the reference section."

Authors' response is

"In the revised part: Page 10 and 15, Revised manuscript

Noticeably, the polarization currents at 800 and 675 mVIR-free are 308 and 1125 mA cm⁻², respectively, which exceeds the United States Department of Energy (DOE) target.⁴⁵ Meanwhile, the hydrogen-air fuel cell works stably at 0.6 V for 24 h (Fig. 5e).

⁴⁵ Myers, D., Kariuki, N., Ahluwalia, R., Wang, X. & Peng, J. K. Rationally Designed Catalyst Layers

for PEMFC Performance Optimization (US DOE, 2015);

https://www.hydrogen.energy.gov/pdfs/review15/fc106_myers_2015_o.pdf

Note that DOE 2020 target for current density at 800 mV is 300 mA/cm². This target is from DIRECT measurement of H₂-air fuel cell WITHOUT any iR correction, as I mentioned in my original comment.

Authors seemed to have difficulty to understand that. Based on their revised text, it is not clear to me if the cell voltage in the Fig. 5d is "iR corrected" or "not corrected"? It would make huge difference.

H₂-air fuel cell measurement should NOT be iR-corrected. The iR correction will make current density and power density artificially high. This is a critical question, which must be clarified before the manuscript can be accepted for publication.

In addition, the current ref. 45 is NOT a proper citation for DOE target. It needs to be replaced by the proper citation of DOE website.

"US DOE Fuel Cell Technology Office Multi-Year Research, Development, and Demonstration Plan

https://www.energy.gov/sites/default/files/2017/05/f34/fcto_myRDD_fuel_cells.pdf,"

which was updated by DOE in 2017. Note Table 3.4.5 and 3.4.7 for MEA and catalyst performance target. Authors are also strongly advised to read the fuel cell test protocol in the appendix.

This manuscript cannot be accepted until these issues are clarified and explained.

Reviewer comments

Reviewer #1 (Remarks to the Author):

I have carefully reviewed authors' response and believed my and other major concerns have been addressed.

There is one minor thing that needs to be corrected before final publishing in Nature Communications.

In Supplementary Table S2, the σ^2 of Co-Pt bond in PtCo@CoNC/NTG is negative, -0.0185, which doesn't make sense. Please double check and reanalyze.

Response: Thanks for your professional suggestions. We carefully recheck the fitting results of Pt L₃-edge and Co K-edge signals for PtCo@CoNC/NTG, paying special attention to the debye-waller factor. Significantly, we change the k-weight = 2, and hope all updated fitting results are satisfactory.

Supplementary Table 1. Structural parameters of Pt foil, PtCo@CoNC/NTG, and PtO₂ extracted from the EXAFS fitting ($S_0^2 = 0.73$).

Sample	Bond	CN	R/Å	ΔE_0 /eV	$\sigma^2/\text{Å}^2$	R-factor
Pt foil	Pt-Pt	12*	2.762 ± 0.003	7.242 ± 0.477	0.0040 ± 0.0003	0.004
PtCo@CoNC/ NTG	Pt-Pt	3.74 ± 1.48	2.691 ± 0.010	5.267 ± 0.864	0.0048 ± 0.0021	0.003
	Pt-Co	5.92 ± 0.70	2.578 ± 0.005		0.0057 ± 0.0008	
PtO ₂	Pt-O	4.84 ± 0.68	2.018 ± 0.011	12.668 ± 1.587	0.0002 ± 0.0015	0.016

Note: S_0^2 is the amplitude reduction factor (obtained by the fitting of Pt foil); CN is the coordination number; R is interatomic distance (the bond length between Pt central atoms and surrounding coordination atoms); ΔE_0 is energy shift; σ^2 is Debye-Waller factor (a measure of thermal and static disorder in absorber-scatterer distances); R-factor is used to value the goodness of the fitting; * the experimental EXAFS fit of metal foil by fixing CN as the known crystallographic value.

For Pt L₃-edge of PtCo@CoNC/NTG, data ranges: $2.0 < k < 12.0 \text{ Å}^{-1}$, $1.0 < R < 3.0 \text{ Å}$. The number of variable parameters is 7, out of total 12.44 independent points. The distance of Pt-Pt is based on the crystal

structure of PtCo.

Supplementary Table 2. Structural parameters of Co foil, PtCo@CoNC/NTG, and CoO extracted from the EXAFS fitting ($S_0^2 = 0.84$).

Sample	Bond	CN	R/Å	ΔE_0 /eV	$\sigma^2/\text{Å}^2$	R-factor
Co foil	Co-Co	12*	2.489 ± 0.003	6.296 ± 0.493	0.0068 ± 0.0004	0.004
PtCo@CoNC/NTG	Co-Co	5.25 ± 0.71	2.498 ± 0.007	7.706 ± 1.343	0.0062 ± 0.0010	0.009
	Co-N	0.77 ± 0.52	2.003 ± 0.044		0.0069 ± 0.0126	
	Co-Pt	1.80 ± 1.39	2.813 ± 0.047		0.0121 ± 0.0085	
CoO	Co-O	4.76 ± 0.95	2.119 ± 0.016	1.356 ± 1.164	0.0080 ± 0.0034	0.010
	Co-Co	14.90 ± 2.80	3.033 ± 0.011		0.0101 ± 0.0018	

Note: S_0^2 is the amplitude reduction factor (obtained by the fitting of Co foil); CN is the coordination number; R is interatomic distance (the bond length between Co central atoms and surrounding coordination atoms); ΔE_0 is energy shift; σ^2 is Debye-Waller factor (a measure of thermal and static disorder in absorber-scatterer distances); R-factor is used to value the goodness of the fitting; * the experimental EXAFS fit of metal foil by fixing CN as the known crystallographic value.

For Co K-edge of PtCo@CoNC/NTG, data range: $2.0 < k < 12.3 \text{ Å}^{-1}$, $1.0 < R < 3.3 \text{ Å}$. The number of variable parameters is 10, out of total 14.81 independent points. The distances of Co-N and CoPt are based on the crystal structure of CoN and PtCo.

Reviewer #2 (Remarks to the Author):

The authors have made revisions on material characterizations and put more details on the electrochemical performance evaluation.

Comment 1. (1) If this paper concludes that PtCo@CoNC/NTG is a potentially good catalyst, the reviewer agrees. The stability demonstrated by this non-PGM catalyst can be of interest to the fuel cell community;

however, the performance on ORR activity and power density are not impressive, as shown by this new table added (Table S4). (2) Given it is a PGM/non-PGM hybrid structure, there could be a potential question on whether the activity and stability performance is attributed to the PGM catalyst. (3) 0.1 mg/cm² PGM loading is sufficient to deliver the activity and stability performance as shown by recent literature (for example Du et al. 10.1149/1945-7111/ac3598; Mauger et al. 10.1016/j.jpowsour.2021.230039).

Response: Regarding the reviewers' concerns, we mainly elucidate the advantages of PtCo@CoNC/NTG catalyst from the following three aspects:

(1) As you mentioned, PtCo@CoNC/NTG is a potentially good catalyst. Compared with the previously reported ORR catalysts, the PtCo@CoNC/NTG catalyst, although not the Number 1 in ORR activity and power density, still exhibits top-level performance for hydrogen-air fuel cell.

(2) We have to reiterate the original intention and potential advantages of the hybrid catalyst design to answer your comment (Given it is a PGM/non-PGM hybrid structure, there could be a potential question on whether the activity and stability performance is attributed to the PGM catalyst). Meanwhile, this is very relevant to the point you mentioned in Comment 2(1), and we respond together here.

For one thing, to overcome the problem of activity decline caused by corrosion, exfoliation and agglomeration of Pt nanocrystals and carbon supports in Pt/C catalysts, we focus on upgrading carbon supports in Pt/C catalysts and develop a PGM/non-PGM hybrid structure to further improve the structural stability and

corrosion resistance. The non-PGM component you mentioned is actually our progress in regulating the graphitization, structure and composition of the carbon support, which is beneficial to improve the corrosion resistance of the carbon support itself. Also, the carbon support coating structure can protect the PtCo alloy and improve its structural stability, thereby enhancing the stability of the entire hybrid catalyst.

For another thing, PtCo alloys and graphitic carbon containing Co-N-C sites of PtCo@CoNC/NTG are significantly upgraded compared to Pt/C. The introduction of Co-N-C increases the number and density of ORR active sites, and the synergistic catalysis of Co-N-C and PtCo can further enhance the ORR catalytic efficiency (Angew. Chem. Int. Ed. 2021, 202115835). More importantly, based on the integrated construction and synergistic catalysis of PtCo alloy and Co-N-C sites, this research work constructs a multi-dimensional continuous carbon network support to optimize the local electron transfer and mass transport during the ORR process, especially in GDE and MEA electrochemical evaluations.

In a word, the enhanced performance of PtCo@CoNC/NTG integrated catalyst benefits from the synergistic promotion of the PtCo alloy and the multi-dimensional continuous carbon network containing Co-N-C sites, including the synergistic catalysis of multiple active sites, the promotion of porous channels, and the protection of the Pt alloy by the carbon-coated structure, etc. The enhanced ORR performance is attributed to the combined effect of the Pt alloy and the modified carbon network, and it is currently difficult to quantify the respective activity and stability contributions of both, which is also a current research difficulty (Angew. Chem. Int. Ed. 2021, 60, 25530; Energy Environ. Sci. 2021, 14, 4948). Hope the above responses satisfy you.

(3) Fuel cell is a systematic project whose performance is not only related to components such as catalytic layers, ionic polymers, proton exchange membranes, and diffusion layers, but also to many test conditions such as humidity, temperature, back pressure, and gas flow rate. Notably, our work mainly focuses on the design and optimization of the cathode catalyst, which shows that there is still room for improvement in other components of the fuel cell. Whereas the recent literatures you mentioned are mainly focus on the design of R2R-coated gas diffusion electrode and the improvement of ionomer distribution (Mauger et al. 10.1016/j.jpowsour.2021.230039), and the effects of dispersion method on catalyst ink's rheological behavior, particle size distribution, catalyst layer's cracks, and performance (Du et al. 10.1149/1945-7111/ac3598), which is clearly different from our research focus. Additionally, we approve of your mention that 0.1 mg/cm^2 PGM loading is sufficient to deliver the activity and stability performance, in compliance with US DOE testing protocols. In our hydrogen-air fuel cell tests, both cathode and anode Pt loadings are 0.1 mgPt/cm^2 and achieve high operating efficiency.

Our work
Anode: 0.1 mg/cm², Cathode: 0.1 mg/cm²

Mauger et al.
Anode : 0.1 mg/cm², Cathode: 0.1 mg/cm²

Du et al.
Anode : 0.1 mg/cm², Cathode: 0.2 mg/cm²

(Mauger et al. 10.1016/j.jpowsour.2021.230039)

(Du et al. 10.1149/1945-7111/ac3598)

Comment 2. (1) If the paper is to study the role of non-PGM catalysts as catalyst support for PGM catalysts, the work lacks convincing evidence. (2) Shown by Figure S7/S8, the PtCo particles have different sizes, is it possible that the authors saw a size-effect? If the authors fit the XAS on the Pt in Figure S18/S19, are there Pt-Co bond changes in CN and distances? Overall, the authors have not presented why PtCo@CoNC/rGO, PtCo@CoNC/CNT, and PtCo@CoNC are not as good as PtCo@CoNC/NTG with experimental evidence. (3) As a result, the claims on the "multi-dimensional architectures and polyatomic active sites collectively strengthen the synergistic ORR catalytic activity and stability" appears groundless.

Response: Taking into account your concerns above, we provide a point-to-point response below:

(1) We have explained in detail the important role of non-PGM component in PGM/non-PGM hybrids in Comment 1(2) above. It should be noted that our focus on efficient ORR hybrid catalysts, not pure PGM or non-PGM catalysts. Graphitic carbon containing Co-N-C not only acts as a support to disperse PtCo nanoparticles, but also contributes some activity to ORR catalysis as an active material, which is a significant

advantage compared to traditional XC-72 carbon support. For our PtCo@CoNC/NTG sample, multidimensional carbon network containing Co-N-C sites can effectively disperse PtCo nanoparticles to increase the utilization of active sites, and the carbon-coated structure can enhance the structural stability of PtCo alloys. Meanwhile, the synergistic catalysis of Co-N-C sites and PtCo alloy reduces the two-electron product and significantly enhances the ORR catalytic efficiency. More significantly, the construction of multi-dimensional carbon network support promotes the local electron transfer and mass transport, especially in GDE and MEA electrochemical evaluations. These results have been verified in the revised manuscript by physical characterization, performance evaluations and theoretical calculations. Therefore, the role of multidimensional graphitic carbon network containing Co-N-C sites in integrated catalysts is critical, not just supports for PGM catalysts.

(2) Based on XAS/TEM analysis, although the size and composition of PtCo nanoparticles are similar, the difference in CN, bond distance and size you mentioned is inevitable, which is difficult to achieve controllable preparation in most high-temperature preparation strategies, let alone on carbon network supports of different dimensions. For a series of PtCo@CoNC integrated catalysts, the intrinsic active sites are mainly composed

of PtCo alloy and graphitic carbon containing Co-N-C groups, and their elemental compositions are also very close, and the difference is mainly reflected in the different carbon network structures.

Notably, we prepare a series of PtCo@CoNC integrated catalysts, not to emphasize the difference of these materials, but to construct four different integrated catalysts following a similar design philosophy. Compared with the PtCo@CoNC, PtCo@CoNC/CNT, and PtCo@CoNC/rGO integrated catalysts, the superior ORR catalytic performance of PtCo@CoNC/NTG is attributed to the advantages of the multidimensional carbon network in specific surface area, pore size distribution and carbon composition (Energy Environ. Sci. 2019, 12, 2830), which is beneficial to strengthen mass transport and active site utilization.

(3) As you mentioned (As a result, the claims on the "multi-dimensional architectures and polyatomic active sites collectively strengthen the synergistic ORR catalytic activity and stability" appears groundless), we have to reiterate the design ideas and potential advantages of PtCo@CoNC/NTG once again to answer your comment. Previously, we have demonstrated the synergistic ORR catalysis of M-N-C sites and Pt alloy can further enhance the reaction efficiency (Angew. Chem. Int. Ed. 2021, 202115835). Based on the integrated construction and synergistic catalysis of PtCo alloy and Co-N-C sites, this work is committed to construct a multi-dimensional continuous carbon network support to optimize the local electron transfer and mass

transport, especially in GDE and MEA electrochemical evaluations. Significantly, the construction of multidimensional carbon network support promotes the mass transport during synergistic ORR catalysis.

In terms of the PtCo@CoNC/NTG integrated catalyst, the multidimensional carbon structure including rGO and NCNT provides a broad platform and abundant channels for electron transfer and gas diffusion, which is beneficial to enhance the ORR catalytic efficiency. The graphitic carbon-coated structure can improve the corrosion resistance and structural stability of the support and Pt alloy, thereby maintaining the efficient catalytic activity of the integrated catalyst. In addition, the synergistic ORR catalysis of PtCo alloy and graphitic carbon network containing Co-N-C active sites reduces the two-electron product, which in turn strengthens the ORR catalytic performance. Consequently, the claims on the "multi-dimensional architectures and polyatomic active sites collectively strengthen the synergistic ORR catalytic activity and stability" is reasonable, which is significantly reflected in material construction and catalytic performance.

Comment 3. The reviewer cannot see the relevance of Fig 6e-h to the study, even after the revision. The scenario seems to be universal, not limited to a single catalyst system. The flow channels and the channels in

the catalyst support are not at the same length scale, some details are required for helping the readers to see how the model applies.

Response: This is a constructive comment for our present work. In our mind, the construction of porous channels in the carbon support network can optimize local oxygen diffusion and water discharge, which is remarkably reflected in the morphological construction and structural optimization of integrated catalysts, GDE and MEA performance enhancements.

As you said, the simulation system seems to be universal, not limited to a single catalyst system. The original purpose of our simulations is simply to show the facilitation of oxygen transport and water discharge by the porous channels design. This physics simulation does not seem to enhance the understanding of the catalytic system, and we decide to remove the finite element simulation so as not to confuse the readers. Hope to get your approval.

In the revised part: Page 20, Revised manuscript

Fig. 6 | Theoretical calculations. **a** Schematic diagram of Pt/C and PtCo@CoNC for synergistic ORR. **b** Free-energy diagram of the ORR pathways for Co-N-C, Pt(111) and PtCo sites. **c, d** Synergistic ORR catalytic

pathways over Co-N-C and PtCo sites at 0 (**c**) and 1.23 V (**d**).

Reviewer #3 (Remarks to the Author):

1. Authors made effort in their revised manuscript. I am generally fine with their responses except to my initial comment 4. My initial comment 4 is as the following:

“Between Line 199 to 201, several statements need to be corrected. DOE target for Pt mass activity in MEA should be measured at 0.9V iR-free, instead of 0.85 V. The current density at 0.8V should be measured directly using the actual cell voltage instead of iR corrected voltage. Also, when use DOE target, correct link to the webpage should be cited in the reference section.”

Authors' response is

“In the revised part: Page 10 and 15, Revised manuscript

Noticeably, the polarization currents at 800 and 675 mV_{iR-free} are 308 and 1125 mA cm⁻², respectively, which exceeds the United States Department of Energy (DOE) target.⁴⁵ Meanwhile, the hydrogen-air fuel cell works stably at 0.6 V for 24 h (Fig. 5e).

45 Myers, D., Kariuki, N., Ahluwalia, R., Wang, X. & Peng, J. K. Rationally Designed Catalyst Layers for PEMFC Performance Optimization (US DOE, 2015);

https://www.hydrogen.energy.gov/pdfs/review15/fc106_myers_2015_o.pdf”

Note that DOE 2020 target for current density at 800 mV is 300 mA/cm². This target is from DIRECT measurement of H₂-air fuel cell WITHOUT any iR correction, as I mentioned in my original comment. Authors seemed to have difficulty to understand that. Based on their revised text, it is not clear to me if the cell voltage in the Fig. 5d is “iR corrected” or “not corrected”? It would make huge difference. H₂-air fuel cell measurement should NOT be iR-corrected. The iR correction will make current density and power density artificially high. This is a critical question, which must be clarified before the manuscript can be accepted for publication.

Response: We very much appreciate your comments that DOE targets should not be IR corrected. In the last revision, we have marked IR-free, indicating that our hydrogen-air fuel cell measurements were not IR corrected. Please check again for traces of our previous revision.

In order to avoid misleading the readers, we illustrate the MEA testing process in more detail in the Supplementary Information to further clarify that our MEA measurements are IR-free, that is, without any IR correction.

In the revised part: Page 10, Revised manuscript

the polarization current at 800 mV_{IR-free} is 308 mA cm⁻², which exceeds the United States Department of Energy (DOE) target.⁴⁵

In the revised part: Page S6, Supplementary information

the hydrogen-air fuel cell measurements were not IR corrected.

2. In addition, the current ref. 45 is NOT a proper citation for DOE target. It needs to be replaced by the proper citation of DOE website.

“US DOE Fuel Cell Technology Office Multi-Year Research, Development, and Demonstration Plan https://www.energy.gov/sites/default/files/2017/05/f34/fcto_myRDD_fuel_cells.pdf,”

which was updated by DOE in 2017. Note Table 3.4.5 and 3.4.7 for MEA and catalyst performance target.

Authors are also strongly advised to read the fuel cell test protocol in the appendix.

This manuscript cannot be accepted until these issues are clarified and explained.

Response: Thanks for your professional comments. Our previously cited reference (Myers, D., Kariuki, N., Ahluwalia, R., Wang, X. & Peng, J. K. Rationally Design Catalyst Layers for PEMFC Performance Optimization, US DOE, 2015) identifies the DOE test protocol and fuel cell performance targets, but this document was updated by the DOE in 2017. We replace it with the correct citation from the DOE website you

mentioned.

In the revised part: Page 14, Revised manuscript

45 US DOE Fuel Cell Technology Office Multi-Year Research, Development, and Demonstration Plan.

https://www.energy.gov/sites/default/files/2017/05/f34/fcto_myrd_d_fuel_cells.pdf

REVIEWER COMMENTS

Reviewer #1 (Remarks to the Author):

I am OK with the revisions that authors made to address my concerns.

Overall, the authors solved the raised questions and clarified their ways to enhance the catalytic stability. However, the author should mention in the revised manuscript that the different catalytic activities of the 4 different PtCo@CoNC samples are due to different carbon specific surface area, pore size distribution and carbon composition, thus avoiding the misunderstanding and confusion. I also suggested authors to cite the related literatures for catalytic activity difference. In addition, the authors should emphasize the stability improvements compared with that using pure carbon as support.

By checking my previous comments again, it seems that in the revised they did not solve the wavelet transfer problem. I do not believe their wavelet transfer results since it lacks Co-O scattering in CoO. They may use wrong parameters. They should delete all wavelet transfer arguments or redo that with higher resolution to argue such Co-N and Co-Pt scattering.

Reviewer #3 (Remarks to the Author):

Authors have made the corrections per my suggestion. The manuscript is acceptable for the publication, except one more correction:

In replying my comment on H₂-air fuel cell measurement should not be iR corrected, Authors replied as the following:

"In order to avoid misleading the readers, we illustrate the MEA testing process in more detail in the Supplementary Information to further clarify that our MEA measurements are IR-free, that is, without any IR correction."

Please note, iR-free = WITH iR correction. It means that the MEA measurement data has adjusted after removing the voltage drop from the to the internal impedance in the polarization curve. Therefore, iR-free data are usually better than the iR-uncorrected one. It is exactly opposite to what Authors understood.

Reviewer #1 (Remarks to the Author):

I am OK with the revisions that authors made to address my concerns.

Additional comments:

1. Overall, the authors solved the raised questions and clarified their ways to enhance the catalytic stability. However, the author should mention in the revised manuscript that the different catalytic activities of the 4 different PtCo@CoNC samples are due to different carbon specific surface area, pore size distribution and carbon composition, thus avoiding the misunderstanding and confusion. I also suggested authors to cite the related literatures for catalytic activity difference. In addition, the authors should emphasize the stability improvements compared with that using pure carbon as support.

Response: As you mentioned, the different catalytic activities of the four different PtCo@CoNC samples are due to different carbon specific surface area, pore size distribution and carbon composition. It should be emphasized that PtCo@CoNC/NTG exhibits improved stability compared to commercial Pt/C, which is attributed to the optimization of the carbon support network. We supplement the description and cite some related literatures, thanks for your valuable comments.

In the revised part: Page 8, 9, Revised manuscript

The different catalytic activities of these four PtCo@CoNC samples are mainly attributed to the differences in carbon specific surface area, pore size distribution, and carbon composition verified above.⁴⁰⁻⁴²

In contrast to using pure carbon as a support, the stability improvement of Pt catalysts by constructing a graphitic nanocarbon network is remarkable.

Reference

40 Qiao, Z. *et al.* 3D porous graphitic nanocarbon for enhancing the performance and durability of Pt catalysts: A balance between graphitization and hierarchical porosity. *Energy Environ. Sci.* **12**, 2830–2841 (2019).

- 41 Lee, Y. J. *et al.* Ultra-low Pt loaded porous carbon microparticles with controlled channel structure for high-performance fuel cell catalysts. *Adv. Energy Mater.* **11**, 2102970 (2021).
- 42 Qiao, Z., Wang, C., Zeng, Y., Spendelov, J. S. & Wu, G. Advanced nanocarbons for enhanced performance and durability of platinum catalysts in proton exchange membrane fuel cells. *Small* **17**, 2006805 (2021).

2. By checking my previous comments again, it seems that in the revised they did not solve the wavelet transfer problem. I do not believe their wavelet transfer results since it lacks Co-O scattering in CoO. They may use wrong parameters. They should delete all wavelet transfer arguments or redo that with higher resolution to argue such Co-N and Co-Pt scattering.

Response: Thanks for your professional suggestions. In the previous wavelet transform of CoO, the Co-O scattering signal can be significantly observed. The relatively weak Co-Pt and Co-N signals are close to Co-Co in distance, so they cannot be accurately shown in the wavelet transform results. In the revised manuscript, the fitted scattering paths can effectively verify various types of coordination signals in PtCo@CoNC/NTG, which correspond to the fitting parameters we provide (Supplementary Table 1, 2). To avoid misleading readers, we delete all wavelet transfer arguments in the manuscript and Supplementary information, and hope these updated revisions are satisfactory.

In the revised part: Page 7 and 8, Revised manuscript

Compared with the k and R space diagrams of Pt foil and PtO₂ (Supplementary Fig. 14), the fitting information presented by PtCo@CoNC/NTG further supports a compressive Pt-Pt distance and an optimized coordination environment (Supplementary Table 1).

Moreover, FT-EXAFS at the Co K-edge show a main peak at ~2.2 Å and some shoulder peaks at ~1.5 and ~2.5 Å (Fig. 3i, Supplementary Fig. 15), which reasonably fit with the scattering paths of Co-Co, Co-N, and Co-Pt, respectively (Supplementary Table 2).

Fig. 3 | Chemical state and coordination environment of PtCo@CoNC/NTG. a-c XPS spectra of N 1s (a), Co 2p (b) and Pt 4f (c). d Pt L₃-edge XANES, e the Fourier transforms of EXAFS spectra, and f the applied scattering paths of Pt L₃-edge for PtCo@CoNC/NTG. g Co K-edge XANES, h the Fourier transforms of EXAFS spectra, and i the applied scattering paths of Co K-edge for PtCo@CoNC/NTG.

Reviewer #3 (Remarks to the Author):

Authors have made the corrections per my suggestion. The manuscript is acceptable for the publication, except one more correction:

In replying my comment on H₂-air fuel cell measurement should not be iR corrected, Authors replied as the following:

"In order to avoid misleading the readers, we illustrate the MEA testing process in more detail in the Supplementary Information to further clarify that our MEA measurements are iR-free, that is, without any IR correction."

Please note, iR-free = WITH iR correction. It means that the MEA measurement data has adjusted after removing the voltage drop from the to the internal impedance in the polarization curve. Therefore, iR-free data are usually better than the iR-uncorrected one. It is exactly opposite to what Authors understood.

Response: This comment is constructive and gives us a better understanding of the DOE testing protocol. We agree with your mention of iR-free = WITH iR correction. Here, we again clarify that the MEA testing process does not perform any iR correction. We revise all inaccurate descriptions of DOE testing protocol in the manuscript and Supplementary information. Thank you very much for your valuable comments.

In the revised part: Page 10, Revised manuscript

PtCo@CoNC/NTG delivers a current density of 1.50 A cm⁻² at 0.6 V_{iR-uncorrected}.

Noticeably, the polarization current at 800 mV_{iR-uncorrected} is 308 mA cm⁻², which exceeds the United States Department of Energy (DOE) target.

REVIEWER COMMENTS

Reviewer #1 (Remarks to the Author):

Authors have addressed my concerns and I am OK with the current version for publishing in Nat Comm.

Reviewer #3 (Remarks to the Author):

Authors have addressed my concerns and made corrections correspondingly. The manuscript can be accepted for publication, in my opinion.